# Pumilio-dependent localization of mRNAs at the cell front coordinates multiple pathways required for chemotaxis

Manuel Hotz[1] & W. James Nelson[1,2]

Chemotaxis is a specialized form of directed cell migration important for normal development, wound healing, and cancer metastasis. In the social amoeba *Dictyostelium discoideum*, four signaling pathways act synergistically to maintain directional cell migration. However, it is unknown how these pathways are coordinated in space and time to achieve persistent chemotaxis. Here, we show that the mRNAs and proteins of these four chemotaxis pathways and actin are preferentially enriched at the cell front during dynamic cell migration, which requires the Pumilio-related RNA-binding protein Puf118. Significantly, disruption of the Pumilio-binding sequence in chemotaxis pathway mRNAs, or mislocalization of Puf118 and its target mRNAs to the cell rear perturbs efficient chemotaxis in shallow cAMP gradients, without affecting the abundance of the mRNAs or encoded proteins. Thus, the polarized localization of Puf118-bound mRNAs coordinates the distribution of different chemotaxis pathway proteins in time and space, leading to cell polarization and persistent chemotaxis.

[1] Department of Biology, Stanford University, Stanford, CA 94305, USA. [2] Molecular and Cellular Physiology, Stanford University, Stanford, CA 94305, USA. Correspondence and requests for materials should be addressed to W.J.N. (email: wjnelson@stanford.edu)

Chemotaxis, a specialized form of directed cell migration important for normal development, wound healing, and cancer metastasis, involves the polarization of the entire cell to drive persistent migration along a chemoattractant gradient[1, 2]. Detailed genetic and cell biological studies over several decades identified a key role for the master regulator Ras in coupling G-protein-coupled receptor-dependent detection of a diffusible chemoattractant gradient to an intracellular signaling pathway for cell polarization and migration[3]. This signaling pathway controls the asymmetric distributions of phosphoinositol (3,4,5)-triphosphate (PIP3) by PI 3-kinase (PI3K) and phosphoinositol (4,5)-bisphosphate (PIP2) by the phosphatase PTEN at the front and rear of the cell, respectively[4, 5], which were thought at the time to be sufficient to maintain directional cell migration by restricting PIP3-dependent activation of Arp2/3 and polymerization of F-actin to the cell front[1, 6–8].

However, recent genetic studies in which all PI3K and PTEN genes were deleted in the social amoeba *Dictyostelium discoideum*[6–8], revealed that natural, cAMP-dependent chemotaxis during *D. discoideum* differentiation required three additional signaling pathways. These pathways acted synergistically with the PI3K pathway in chemotactic signal amplification and memorization to maintain directional cell migration in a shallow, physiological gradient[9, 10]: TorC, the phospholipase Pla2, and the guanylyl cyclases SgcA and GcA. All three pathways, and the PI3K pathway, were shown to act downstream of initial Ras-dependent cell polarization, and were required to maintain and stabilize directed chemotactic migration[9, 10]. These four different signaling pathway proteins have no obvious homologies or binding partners, and hence, it is unknown how they are localized in time and space during dynamic cell migration.

The goal of this study was to investigate whether there is a common mechanism for coordinating the localization of these pathways at the cell front during directed chemotactic cell migration. We show that proteins of these four chemotaxis pathways are all localized to the cell front, and that their distribution is dependent on the mRNA-binding protein Puf118 that binds and localizes their mRNAs to the front of migrating cells. Mutation of the 3′-UTR Puf118 binding site in chemotactic pathway mRNAs, or mislocalization of Puf118 and these mRNAs to the cell rear inhibits chemotaxis. Thus, Puf118-dependent localization of chemotaxis pathway mRNAs to the cell front is a

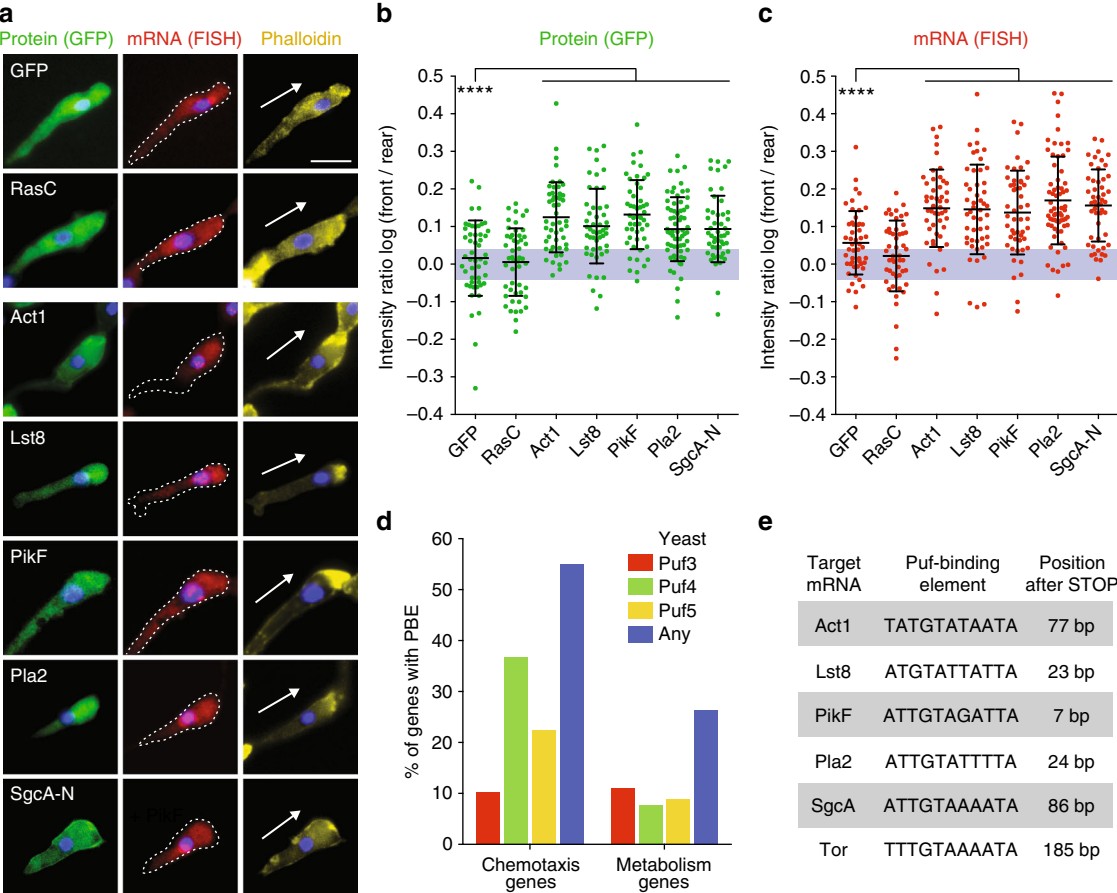

**Fig. 1** Four chemotaxis pathway mRNAs and proteins localize to the cell front in chemotaxis. **a** Localization of GFP-tagged proteins and corresponding mRNAs in cells expressing GFP-RasC, -Act1, -Lst8, -PikF, -Pla2, the N-terminal 1019 residues of SgcA (SgcA-N) and GFP control in natural chemotactic streams. Chemotaxis pathway genes were expressed with their endogenous 3′-UTRs. Arrow indicates orientation of cell polarity defined by F-actin stain (Alexa Fluor 647-conjugated Phalloidin). DNA (nucleus) was stained with DAPI. Note that some GFP-mRNAs were also localized in bright foci within the nucleus, which represent sites of transcription. Scale bar, 10 μm. **b**, **c** Quantification of the fluorescence intensity ratio of each GFP-tagged protein and mRNA between the front and rear of the cell, and represented as a log of the log F/R. The gray area indicates values equivalent to symmetric localization (between log(0.9) and log(1.1)). Mean and standard deviation (SD), $n \geq 50$ cells. Mann–Whitney test: ****$p < 0.0001$. **d** Percentage of genes annotated with the function "chemotaxis" (97 genes) or "metabolism" (57 genes) in Dictybase with PBEs defined by consensus sites for yeast Puf3, Puf4, Puf5 or any of them within 330 base pairs after STOP codon (Supplementary Fig. 2). **e** Pumilio-binding elements (PBEs) in the 3′-UTR of genes involved in chemotaxis

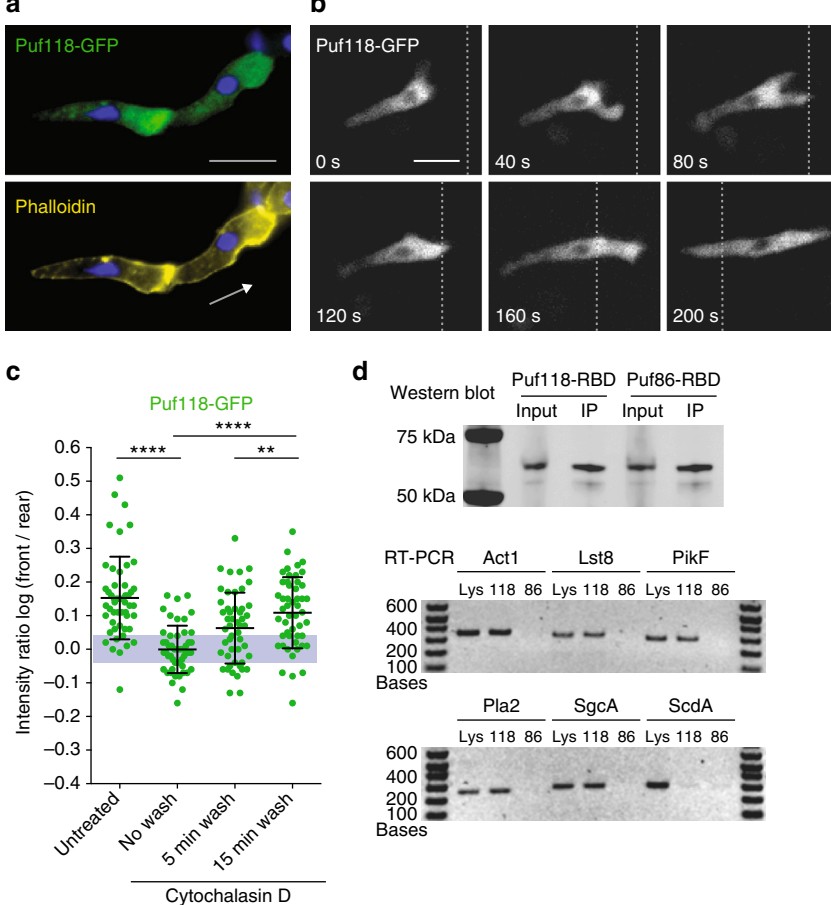

**Fig. 2** The RNA-binding protein Puf118 localizes to the cell front and binds chemotaxis pathway mRNAs. **a** Polarized localization of Puf118-GFP to the cell front in cells migrating in a natural chemotaxis gradient. Arrow indicates orientation of cell polarity defined by F-actin stain (Alexa Fluor 647-conjugated Phalloidin). DNA (nucleus) was stained with DAPI. Scale bar, 10 μm. **b** Time-lapse microscopy on Puf118-GFP in a cell migrating in a natural chemotaxis gradient, relative to a fiducial mark (dotted line). Scale bar, 10 μm. **c** Puf118-GFP log F/R before (untreated), during (no wash) and after (wash) incubation with Cytochalasin D. The gray area indicates values equivalent to symmetric localization (between log(0.9) and log(1.1)). Mean and SD, $n \geq 50$ cells. Mann–Whitney test: ****$p < 0.0001$, **$p < 0.01$. **d** Coimmunoprecipitation (IP) of GFP-Puf118-RBD ("118") and GFP-Puf86-RBD ("86") with a GFP antibody, and immunoblotted with the GFP antibody (western blot) or processed for RT-PCR for the indicated mRNAs (RT-PCR). "Lys" is input lysate control

common mechanism that colocalizes and coordinates signaling from these pathways for efficient chemotaxis in a physiological chemoattractant gradient.

## Results

**Chemotaxis-related proteins and mRNAs are at the cell front**. Using GFP-tagged proteins expressed under control of their endogenous promoter, we showed that Lst8-GFP (a TorC pathway component), PikF-GFP, Pla2-GFP, and SgcA-N-GFP all accumulated at the cell front with a distinct front–rear polarization measured as the Log front/rear fluorescence intensity ratio (log F/R~0.1), similar to Act1-GFP and phalloidin-stained F-actin (Fig. 1a, b). For SgcA-N-GFP, this pattern has been reported previously[11]. In contrast, RasC, the upstream master regulator of cell polarization[12], was localized diffusely throughout the cell[3, 13] (log F/R~0), similar to a cytoplasmic GFP control (Fig. 1b). Live-cell imaging of Pla2- and Lst8-GFP showed that the asymmetric distribution of these proteins at the cell front persisted in new leading pseudopods during dynamic cell migration (Supplementary Fig. 1a, b).

Since Lst8, PikF, Pla2, and SgcA lack obvious protein sequence homologies that could explain their colocalization, we examined their mRNA distributions using RNA fluorescence in situ hybridization (FISH). Significantly, mRNAs of GFP-tagged Lst8, PikF, Pla2, and SgcA-N were enriched in the cytoplasm immediately behind the actin-rich cell front during chemotaxis (log F/R~0.15), similar to the encoded proteins (Fig. 1a–c). Act1 mRNA also accumulated at the cell front (Fig. 1a–c), as reported previously in migrating fibroblasts[14, 15]. In contrast, RasC and cytoplasmic GFP mRNAs were distributed diffusely throughout the cell, similar to the encoded proteins (log F/R~0). These and other results (see below) indicate that the asymmetric colocalization of these four chemotaxis pathways and Act1 mRNAs and encoded proteins at the cell front was not due to overestimation of asymmetry as a result of a geometric effect.

**Chemotaxis pathway mRNAs are enriched for PBEs**. In order to identify the mechanism underlying the polarized localization of these mRNAs, we searched the 3′-untranslated region (3′-UTR) of the chemotaxis pathway and actin mRNAs for binding sites of known RNA-binding proteins. We did not find any classical zip-code motifs[16–18] in the four chemotaxis pathway mRNAs (for details see "Methods" section), which had been identified in mammalian β-actin[14] and Arp2/3 mRNAs[19]. However, a genome-wide search for Pumilio-binding elements (PBEs), using the well-defined consensus sites of yeast Puf proteins[20]

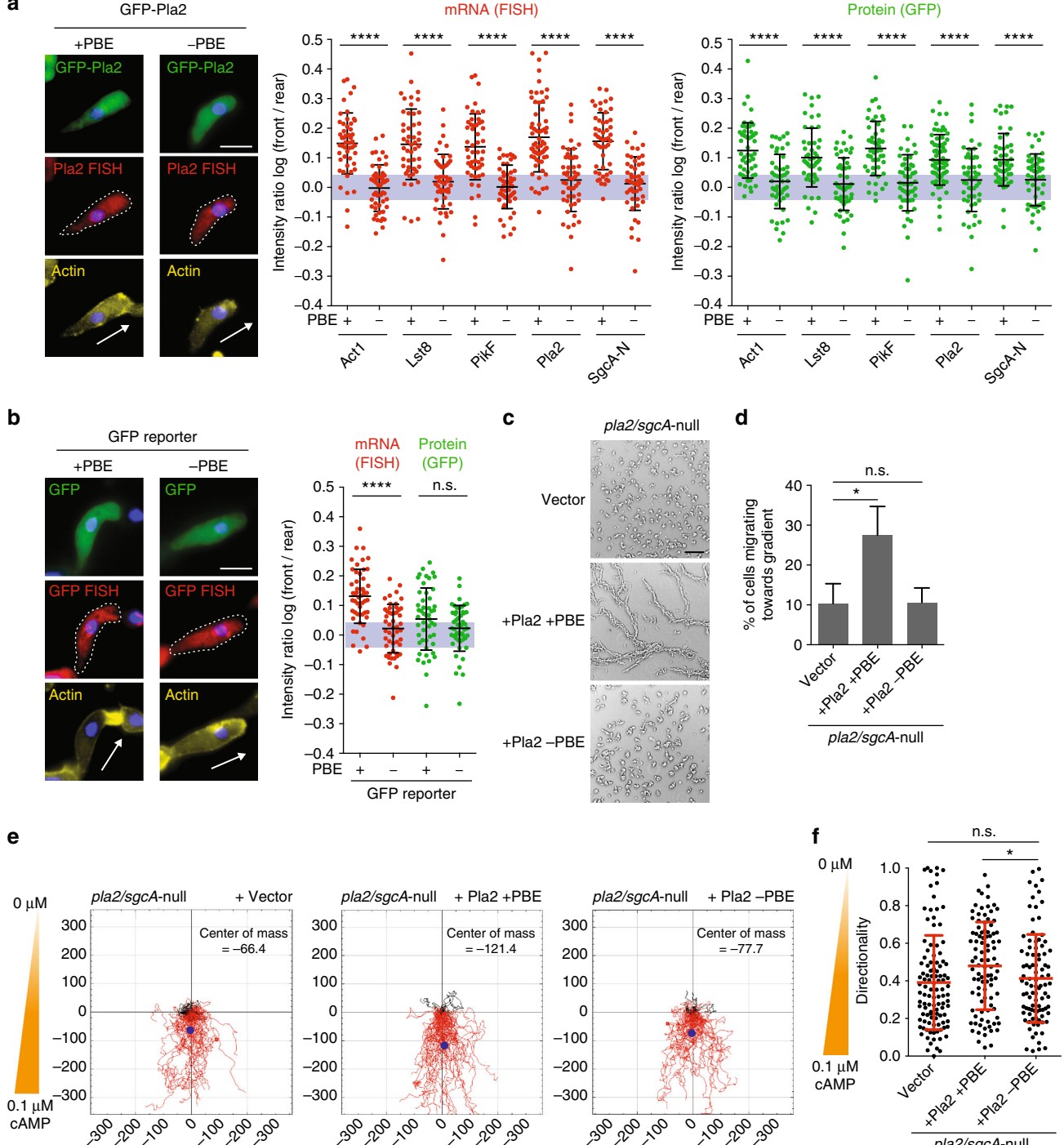

**Fig. 3** The PBE of chemotaxis pathway mRNAs is required for mRNA and protein localization to the cell front and for chemotaxis. **a** Localization and quantified log F/R of GFP-tagged chemotaxis pathway proteins and corresponding mRNAs in cells in natural chemotactic streams. Genes were fused to their endogenous 3′-UTRs with wild-type (+) or mutated (−) PBE. Arrow indicates orientation of cell polarity defined by F-actin stain (Alexa Fluor 647-conjugated Phalloidin). DNA (nucleus) was stained with DAPI. The gray area indicates values equivalent to symmetric localization (between log(0.9) and log(1.1)). **b** Same quantification as in **a** on cells expressing a GFP reporter construct fused to Pla2 3′-UTR (±PBE). For **a** and **b**: mean and SD, $n \geq 50$ cells; Mann–Whitney test: ****$p < 0.0001$; scale bar, 10 μm. **c** Chemotactic streams of *pla2/sgcA*-null cells expressing vector control, Pla2 +PBE or Pla2 -PBE after starvation for 15 h. Scale bar, 300 μm. **d** Percent of *pla2/sgcA*-null cells expressing vector control, Pla2 +PBE or Pla2 -PBE migrating in a chemotaxis chamber for 180 min towards a 0.1 μM/mm cAMP gradient (see also Supplementary Fig. 7C). Mann–Whitney test: *$p < 0.05$, n.s. not significant. **e** Tracks of individual migrating *pla2/sgcA*-null cells expressing vector control, Pla2 +PBE or Pla2 -PBE projected into the same starting point, and center of mass for all cells (blue dot); $n \geq 95$ cells from ≥3 independent experiments. **f** Quantification of the directionality of cells in **e**. Mean and SD; Mann–Whitney test: ****$p < 0.0001$, n.s. not significant

(Supplementary Fig. 2A), recovered a high percentage of genes that were annotated in Dictybase as functioning in "Chemotaxis" (Fig. 1d, e and Supplementary Fig. 2B–D; note that 27 of 31 Act genes, and 6 of 8 PI3K genes in *D. discoideum* contained a PBE; see Supplementary Fig. 2e, f, and Supplementary Note 1). As a control for genes unrelated to chemotaxis, we used the annotation "Metabolism" and "Mitochondria", neither of which were enriched for PBE-containing 3′-UTRs (Fig. 1d, and Supplementary Fig. 2G).

**Puf118 binds chemotaxis-related mRNAs at the cell front**. The PBE is recognized by the RNA-binding domain (RBD) of Pumilio/Puf family proteins, which are involved in mRNA localization, translation suppression and activation[21, 22]. The PBE recognized by yeast Puf4 was the most abundant among *D. discoideum* "Chemotaxis" genes (Fig. 1d, Supplementary Table 1, Supplementary Fig. 2G). Assuming that the specificities of yeast and *D. discoideum* Puf proteins are comparable, we searched for a Puf4-related protein in *D. discoideum*. We found five Puf-related genes (Supplementary Fig. 3a, b; Supplementary Note 1), of which the closest homolog of yeast Puf4 is the previously uncharacterized gene DDB_G0289987, which we refer to as Puf118 based on its predicted molecular mass. The RBD of Puf118 was very similar to that of Puf4 (46% identical, and 64% similar residues), and the residues known to be responsible for mRNA target recognition were nearly identical (90% identical, and 96% similar residues (Supplementary Fig. 3c))[23].

Significantly, GFP-tagged Puf118 localized in the cytoplasm immediately behind the actin-rich cell front during chemotaxis (log F/R~0.15; Fig. 2a–c), similar to the distribution of chemotaxis pathway mRNAs (Fig. 1a, b), and this localization also persisted during dynamic extension and retraction of the pseudopod during cell migration (Supplementary Fig. 4). Puf118-GFP polarization depended on F-actin polymerization at the cell front, since a low dose of cytochalasin D (CD), which disrupted the actin cytoskeleton without causing cell rounding[24], resulted in loss of Puf118-GFP-polarized distribution (log F/R~0; Fig. 2c). However, Puf118-GFP polarization at the cell front was restored upon CD washout concomitant with F-actin reassembly at the cell front (Fig. 2c). Thus, Puf118 localizes at the cell front during directed cell migration, which requires a polarized F-actin cytoskeleton in the leading pseudopod.

We next tested whether chemotaxis pathway mRNAs were bound to Puf118. GFP-Puf118-RBD and, as a control, the related Puf protein GFP-Puf86-RBD (DDB_G0279557; Supplementary Fig. 3) were expressed in cells, immunoprecipitated with an anti-GFP antibody, and coprecipitated mRNAs were identified by RT-PCR (Fig. 2D). The mRNAs of Lst8, PikF, Pla2, SgcA, and Act1, but not the metabolic enzyme ScdA, coimmunoprecipitated with Puf118-RBD, but not with Puf86-RBD.

**PBE is required for localization of mRNAs at the cell front**. To establish that Puf118 directly localized chemotaxis pathway mRNAs in cells, we examined the distribution of mRNAs and proteins with a normal (+)PBE, or a mutated (−)PBE that contained two mutated base pairs (Fig. 3a). All the (+)PBE mRNAs and proteins had a polarized distribution (log F/R~0.15 and ~0.1, respectively), whereas all of the (−)PBE mRNAs and encoded GFP-tagged proteins were diffusely distributed in the cytoplasm (log F/R~0). Importantly, mRNA and protein levels of all (+)PBE and (−)PBE variants were comparable, indicating that the PBEs do not control mRNA abundance or translation, but are primarily required for mRNA localization (Supplementary Fig. 5A–C). Thus, Puf118 binds to and colocalizes chemotaxis pathway mRNAs and Act1 mRNA at the cell front, and this interaction

promotes the polarized localization of the encoded proteins in cells undergoing natural chemotaxis.

To test whether the PBE recognized by Puf118 was sufficient to localize mRNA at the cell front, the 3′-UTR of Pla2 containing a normal (+) or mutant (−)PBE was fused to GFP (Fig. 3b). GFP (+)PBE mRNA localized to the cell front (log F/R~0.13), but GFP (−)PBE mRNA was distributed diffusely in the cytoplasm (log F/R~0). Thus, the PBE is necessary and sufficient for Puf118-dependent localization of the mRNA at the cell front. Interestingly however, GFP protein encoded by (+)PBE GFP mRNA was not enriched at the cell front (Fig. 3b), in contrast to GFP-tagged chemotaxis pathway proteins (Fig. 1a), indicating that chemotaxis pathway proteins are retained at the site of mRNA translation. The mechanism involved in the retention of locally translated chemotaxis pathway proteins at the cell front is unknown (see "Discussion"). A possibility that proteins remain associated with the Puf118 that bound their mRNA seems unlikely since RNA-binding proteins are usually absent from their mRNAs, once the translation machinery is loaded[25].

**PBE-localized mRNAs are required for chemotaxis**. PI3K, Pla2, SgcA, and TorC/Lst8 act synergistically in chemotaxis, since the absence of two or more of these pathways perturbs cell migration in a shallow chemoattractant gradient (e.g., *pla2/sgcA*-null)[9, 10]. Following this logic, we tested whether diffusely localized Pla2 protein from (−)PBE mRNA expressed in *pla2/sgcA*-null cells could rescue chemotaxis (Fig. 3c). While *pla2/sgcA*-null cells expressing Pla2 protein from (+)PBE mRNA formed characteristic streams in a native cAMP gradient, *pla2/sgcA*-null cells with a comparable level of Pla2 protein from (−)PBE mRNA did not and remained in small cell aggregates (Fig. 3c). Chemotaxis of these cells was also measured under controlled conditions in a chemotactic chamber (Supplementary Fig. 6). In a shallow 0.1 µM/mm cAMP gradient that mimics natural chemotaxis, almost three times more *pla2/sgcA*-null cells expressing Pla2 protein from (+)PBE mRNA entered the gradient than *pla2/sgcA*-null cells or *pla2/sgcA*-null cells expressing Pla2 protein from (−)PBE mRNA (Fig. 3d). As expected, chemotaxis by cells expressing Pla2 protein from (+)PBE mRNA was less efficient than wild-type cells (Fig. 4c), indicating a partial rescue due to the remaining *sgcA*-null mutation. Given these phenotypes, we analyzed chemotaxis within a 0.1 µM/mm cAMP gradient in a chemotactic chamber in more detail. The entire population of *pla2/sgcA*-null cells expressing Pla2 protein from (−)PBE mRNA migrated less far into the gradient than *pla2/sgcA*-null cells expressing Pla2 protein from (+)PBE mRNA (Fig. 3e, center of mass), and this was due to a slight, but significant decrease in their ability to maintain a stable and straight trajectory (termed directionality: Fig. 3f, see also Supplementary Fig. 7A). Thus, expression of Pla2 protein from (−)PBE mRNA, which localizes symmetrically in the cytoplasm, blocked the ability of the cells to migrate persistently and maintain direction in a shallow chemoattractant gradient.

The importance of correct Puf118-dependent mRNA localization at the cell front for chemotaxis was tested stringently by mislocalizing Puf118 to the rear of migrating cells. A fragment of Talin A, which localizes to the rear uropod of chemotacting cells[26], was fused to Puf118-RBD. The resulting chimeric protein TalA-Puf118 also localized at the cell rear (Supplementary Fig. 8A), although its localization was more punctate than reported for TalA-GFP[26], possibly due to the presence of the Puf118-RBD or the level of expression. mRNAs of Lst8, PikF, Pla2, SgcA-N, and Act1 all colocalized with TalA-Puf118 at the cell rear in a PBE-dependent manner (Fig. 4a, Supplementary Fig. 8B, C), while their mRNA and protein levels were unchanged by expression of TalA-Puf118 (Supplementary Fig. 5A–D)

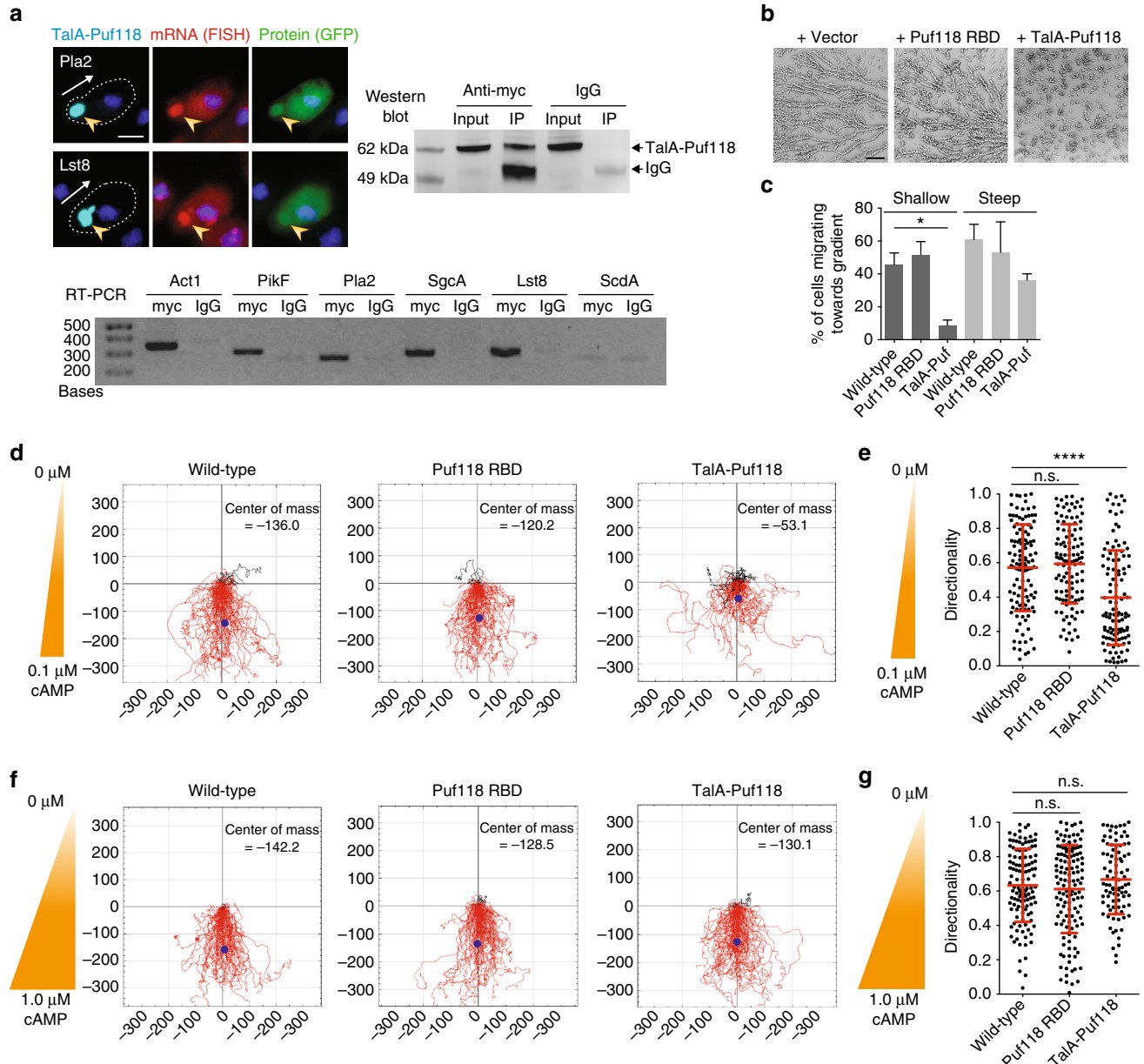

**Fig. 4** Mis-localization of Puf118 to the rear of cells inhibits chemotaxis. **a** Colocalization of GFP-Pla2 and GFP-Lst8 and corresponding mRNAs with myc-tagged TalA-Puf118 (arrowheads) in natural chemotactic streams. Arrow indicates orientation of cell polarity defined by TalA-Puf118 localization. Scale bar, 5 μm. Myc western blot of myc IPs from lysates of cells expressing TalA-Puf118. RT-PCR of indicated mRNAs coimmunoprecipitated with myc-TalA-Puf118. Mouse IgG was used as an immunoprecipitation control. **b** Chemotactic streams of cells expressing vector control, Puf118-RBD or TalA-Puf118 after starvation for 15 h. Scale bar, 300 μm. **c** Percent of wild-type cells, cells expressing Puf118-RBD or TalA-Puf118 migrating towards a 0.1 μM/mm (shallow) or 1 μM/mm (steep) cAMP gradient during 180 min. Mean and SD of $n \geq 3$ experiments; Mann–Whitney test: $*p < 0.05$. **d–g** Tracks (**d**, **f**) and directionality (**e**, **g**) of wild-type cells, cells expressing Puf118-RBD or TalA-Puf118 migrating in a chemotaxis chamber for 180 min in a 0.1 μM/mm (shallow; **d**; **e**) or 1 μM/mm (steep; **f**; **g**) cAMP gradient. $n \geq 110$ cells from $\geq 3$ independent experiments. Mann–Whitney test: $****p < 0.0001$, n.s. not significant

Immunoprecipitation of TalA-Puf118 followed by RT-PCR showed that chemotaxis pathway mRNAs, but not the control ScdA, were in a complex with TalA-Puf118 (Fig. 4a). Not only the mRNAs, but also the encoded GFP-tagged proteins accumulated with TalA-Puf118 (Fig. 4a and Supplementary Fig. 8B, C), indicating that the mislocalized chemotaxis pathway mRNAs at the cell rear were actively translated and the encoded proteins were retained at that site where F-actin was also enriched.

TalA-Puf118 expressing cells did not form streams in a natural, shallow cAMP gradient compared to wild-type or Puf118-RBD control cells (Fig. 4b), and were strongly impaired in their ability to enter (Fig. 4c), and migrate efficiently within (Fig. 4d and

Supplementary Fig. 7B) a shallow cAMP gradient (0.1 μM/mm cAMP) in chemotactic chambers. Importantly, detailed analysis of chemotaxis within the gradient revealed that the directionality of migration was affected as TalA-Puf118 expressing cells failed to maintain persistent migration (Fig. 4e). In contrast, chemotaxis of TalA-Puf118 expressing cells in a ten-fold steeper cAMP gradient (1 μM/mm cAMP), which is independent of the four downstream amplifying pathways and strictly dependent on RasC[10], was normal and similar to control cells (Fig. 4c, f, g and Supplementary Fig. 7C). These results demonstrate that Puf118-dependent localization of four chemotaxis pathway mRNAs and Act1 at the cell front is required for efficient chemotaxis in a

natural, shallow cAMP gradient, when signal amplification and maintenance by the TorC, Lst8, PikF, and Pla2 pathways are required for directed cell migration.

## Discussion

The goal of this study was to investigate whether there is a common mechanism for coordinating the localization of four chemotaxis pathway proteins (PI3K, Pla2, SgcA, and TorC/Lst8) at the cell front for directed cell migration. Our results showed that proper chemotaxis in a shallow, physiological chemoattractant gradient requires Puf118-dependent colocalization of all four chemotaxis pathway mRNAs at the cell front, since expressing (+)PBE, but not (−)PBE Pla2-GFP mRNA rescued chemotaxis of *pla2/sgcA*-null cells (Fig. 3), and TalA-Puf118 mislocalization of chemotactic pathway mRNAs at the cell rear (Fig. 4) inhibited chemotaxis.

However, not only is the localization of chemotaxis pathway mRNAs at the cell front required for chemotaxis, but the subsequent localization/retention of PI3K, Pla2, SgcA and TorC/Lst8 protein at the cell front is also required. For example, the diffuse distribution of (−)PBE Pla2-GFP mRNA and correspondingly Pla2-GFP protein did not rescue chemotaxis in *pla2/sgcA*-null cells (Fig. 3). This indicates that although the Pla2-GFP protein translated from the (−)PBE Pla2-GFP mRNA is the same as that translated from the (+)PBE Pla2-GFP mRNA, it does not function in the chemotaxis pathway because it is not concentrated at the cell front. It is likely that chemotaxis pathway proteins are specifically localized after synthesis at the cell front, since GFP protein is diffuse even though (+)PBE GFP mRNA is localized at the cell front (Fig. 1a–c). These chemotaxis pathway proteins may bind to proteins associated with the actin cytoskeleton at the cell front[27], undergo localized post-translational modifications at the cell front that increases their binding affinity to those proteins, or form complex protein assemblies in biomolecular condensates[28] perhaps mediated by locally high concentrations of F-actin in pseudopods; further studies will be required to test these, and other mechanism(s).

mRNA localization has emerged as a mechanism to break symmetry in migrating fibroblasts[15, 29, 30], but it remains poorly understood how polarization of the entire cell is achieved during persistent migration. Our results build upon, and significantly broaden the functional importance of mRNA localization in cell migration[14, 19] by showing that mRNAs for four chemotaxis signaling pathways (TorC, Lst8, PikF, and Pla2) were coordinated in space and time by binding to Puf118, which localized to the cell front in an F-actin dependent manner. Importantly, persistent cell migration in a natural, shallow cAMP gradient was dependent on the polarized distribution of Puf118 and these mRNAs. Moreover, we showed that Puf118-bound mRNAs colocalized with their encoded proteins (Fig. 4a), indicating that Puf118 localized actively translating mRNA; this is similar to the role of Puf2 in localizing mRNAs of polarity factors to sites of *Ashbya gossypii* polarization[22]. Our model also supports the "mRNA operon" hypothesis which posits that the translation of multiple mRNAs can be coregulated in space and time[31]. Thus colocalization of multiple mRNAs by a common RNA-binding protein coordinates multiple pathways required for directed cell migration in a natural chemoattractant gradient.

Our results indicate that the colocalization of F-actin and Puf118 at the cell front is mutually dependent: Puf118 polarization required localized F-actin assembly (Fig. 2c), and actin mRNA localization in turn depended on Puf118 polarization (Figs. 1, 3). We suggest that F-actin and Puf118 form a positive feedback loop (Supplementary Fig. 9), in which F-actin polymerization at the cell front is initiated by local activation of RasC.

This results in the recruitment of Puf118-bound chemotaxis pathway and Act1 mRNAs, as well as the encoded proteins, which locally amplify and stabilize signals from a weak chemoattractant gradient to maintain localized actin assembly and persistent cell migration. Finally, it is intriguing to speculate that other cells undergoing chemotaxis such as neutrophils might utilize a related mechanism to reliably polarize their motility machinery, since chemotaxis pathways are largely conserved from amoeba to higher metazoans[32].

## Methods

**Cell culture, development, and transformation.** *Dictyostelium discoideum* wild-type Ax2 strain was obtained from Dictybase Stock Center (www.dictybase.org); *pla2/sgcA*-null cells and the corresponding wild-type Ax3 strain for these cells were a gift from A. Kortholt (Univ. of Groningen, Netherlands). Cells were maintained in HL-5 medium using standard culture methods[33].

Cell transformation was performed by electroporation with a BioRad Gene Pulser Xcell as described previously[34]. After transformation, strains were grown in HL-5 medium with the appropriate antibiotics: G418 at 50 µg/ml (Gold Biotechnology, St. Louis, USA), Hygromycin B at 50 µg/ml and Blasticidin C HCl at 10 µg/ml (Thermo Fisher Scientific, Waltham, USA). Cells expressing GFP-tagged chemotaxis pathway proteins were enriched by FACS-sorting to obtain a population with sufficient expression.

Two methods were used to develop cells until chemotaxis occurred: (1) To form natural chemotactic streams, cells were grown on cover slips in cell culture dishes. HL-5 medium was replaced with Development Buffer (DB; 5 mM KH$_2$PO$_4$, 5 mM Na$_2$HPO$_4$, 2 mM MgCl$_2$, 2 mM CaCl$_2$, pH 6.5), and cells were starved overnight. Once streams had formed, cells were fixed (1.5% formaldehyde in 100% ethanol) for 1 h and processed further for imaging (see below); (2) to study chemotaxis in controlled cAMP gradients, cells grown in HL-5 medium were washed three times and resuspended in DB. The cell suspension was pipetted onto 25 mm Nuclepore filters (Whatman) on filter paper soaked with DB, at a density of $7.25 \times 10^6$ cells per filter. After 6 h of starvation, cells were harvested and placed into chemotaxis chambers (see below).

**Plasmid construction.** Expression constructs for GFP-tagged proteins were prepared by homologous recombination in yeast using genomic DNA fragments containing the respective *Dictyostelium* genes (PCR fragment size 1–1.5 kb) and the yeast vector pRS314 (a gift from W. Vonk, Stanford, USA). This approach allowed the assembly of larger genes even when the content of repetitive sequences was extraordinarily high. For Act1, Lst8, PikF, Pla2, and SgcA, 300–600 bp were added following the STOP codon to include the endogenous 3′-UTR; the entire gene was then sub-cloned into a pDM358 vector containing an N-terminal GFP using XbaI and HindIII. Puf118-GFP was sub-cloned into a pDM358 vector (Dictybase) containing C-terminal GFP with BglII and SphI, and GFP-Puf118-RBD and GFP-Puf86-RBD were sub-cloned into the pDM317 vector (Dictybase) using BglII and SpeI.

The GFP reporter construct containing the Pla2 3′-UTR was constructed by adding an extra XbaI site on the GFP-Pla2 pDM358 vectors (+PBE site and -PBE site) immediately next to the STOP codon by site-directed mutagenesis. The Pla2 ORF was cut out, and the remaining vector containing only the GFP and the Pla2 3′-UTR was re-ligated.

The myc-TalA-Puf118 construct was assembled in the pRS314 vector by homologous recombination of three PCR fragments in yeast: 3xmyc amplified from pKT3M (a gift from P. Morgado Flores), the I/LWEQ actin-binding domain of Talin A (base pairs 7060–7660 of the genomic sequence) and the Puf domain of Puf118 (base pairs 2109–3188 of the genomic sequence). The combined insert was then sub-cloned to the pDM304 vector using SpeI and HindIII.

**Chemotaxis experiments.** The µ-Slide Chemotaxis chamber (ibidi, Martinsried, Germany) was used for quantitative chemotaxis experiments. Developed cells were pipetted into the seeding chamber and then the rest of the chamber was filled with DB (Supplementary Fig. 6A). Then, a 10× solution of the desired cAMP concentration (dissolved in DB) was loaded in the chamber opposite the seeding chamber (Supplementary Fig. 6A). Cells were allowed to settle for 30 min before being imaged every 2 min for 4 h by Phase Contrast microscopy using a ×5 objective (NA 0.16). A detailed description of the microscope is given below. This setup differs from the manufacturer's recommendation, since cells are loaded into one of the big reservoirs ("seeding chamber"), rather than the narrow central reservoir ("observation area"). This setup is ideal for imaging and tracking of fast migrating cells like *Dictyostelium*, since cells are allowed to travel the entire distance from one reservoir to the other (1 mm).

**Fluorescence microscopy and sample preparation.** Cells expressing GFP-tagged proteins were induced to form chemotactic streams and fixed as described above. Cell were stained for F-actin with Alexa Fluor 647-conjugated Phalloidin in PBS (1:200 dilution; Thermo Fisher Scientific, Waltham, USA) for 10 min, and washed

three times with PBS before mounting with Vectashield containing DAPI (Vector Laboratories, Burlingame, USA). For RNA-FISH, fixed cells were re-hydrated with Stellaris RNA FISH Wash Buffer A (Biosearch Technologies, Novato, USA) for 20 min, and then hybridized overnight with Stellaris RNA FISH Hybridization Buffer containing 250 nM of FISH probe in a humidified chamber at room temperature. The FISH probe was designed to target the mRNA of GFP, which was codon-optimized for *Dictyostelium discoideum*, and consisted of 24 pooled short RNA probes labeled with CAL Fluor Red 590 Dye (Biosearch Technologies, Novato, USA). Stained cells were washed for 30 min with Wash Buffer A and then stained with Phalloidin as described above. Finally, cells were washed for 5 min with Wash Buffer B and mounted.

Protein immunofluorescence combined with RNA-FISH staining (IF/FISH) was performed by re-hydrating fixed cells in RNase-free PBS for 20 min. Cells were blocked with blocking/staining buffer (2% BSA, 50 mM NH$_4$Cl, 1% donkey serum, 1% goat serum and 1 µl/ml RNaseOut (Thermo Fisher Scientific, Waltham, USA) in RNase-free PBS) for 10 min, and then stained with anti-myc antibody (1:1000 dilution; Roche, Switzerland). After three washes in PBS, Cy5-conjugated anti-mouse secondary antibody (1:1000 dilution; Jackson Immuno Research, West Grove, USA) was added for 1 h, the cells were washed three times, and re-fixed with 1.5% formaldehyde in PBS for 10 min. After two additional washes with Wash Buffer A, samples were processed for RNA-FISH as described above.

All images were taken with a Zeiss Axiovert 200 M inverted wide field epifluorescence microscope (Intelligent Imaging Innovations (3i), Denver, USA) and a Hamamatsu C11440 Digital camera (Orca-flash4.0OLT). The microscope is equipped with a motorized stage and harmonic drive z-focusing, an X-Cite 120 LED illumination source and a quad filter cube for DAPI, FITC, Cy3 and Cy5. Imaging was done with a ×63 objective (NA 0.75) using Slidebook software (3i). Live-cell time-lapse imaging of Puf118-, Pla2-, and Lst8-GFP was done in natural streams, formed as described above, and were imaged every 20 s.

**Image processing and quantifications**. All image analysis was performed using Fiji software. Images of GFP-tagged proteins and RNA-FISH staining were background-subtracted using the extracellular background as a reference. To obtain the front–rear intensity ratio, the mean fluorescence intensity in the pseudopod (the area of the cell between the nucleus and the leading edge) and the mean intensity in the rear of the cell (between the nucleus and the rear end) were measured. Identical areas were selected in both GFP- (protein) and Cy3-channels (FISH). The data are presented as the log of the ratio between front and rear. Due to variations in cell morphology, migration behavior, and mRNA/protein expression levels, only cells within a chemotactic stream with clearly polarized F-actin (phalloidin) and with visible levels of GFP and mRNA were included in the quantification.

Images from chemotaxis chambers were cropped to 1.95 × 0.95 mm to show only the observation chamber, and time-lapse movies were edited to show frame 20–110 (7–250 min after cAMP addition) when optimal chemotaxis occurred in wild-type cells. Images were then background-subtracted (rolling ball radius = 5), and adjusted for brightness and contrast. Next, the images were smoothed, thresholded and inverted. The resulting image was then scaled to fourfold its original size and then reduced back to its original size while averaging; this last step reduced the likelihood of creating symmetric objects which cause erroneous tracking. Cells were then tracked using the TrackMate Plugin using a LoG detector with an Estimated Blob Diameter of 10 and a threshold value of 1, and the Simple Lap Tracker (Linking Max distance = 45, Gap-closing Max distance = 45 and Gap-closing Max frame gap = 3). The Chemotaxis Plugin was used to display the tracks of individual cells, and to quantify the Directionality of the tracks, a measure for the cell's ability to maintain a stable and straight trajectory (http://ibidi.com/software/chemotaxis_and_migration_tool/). We also determined the forward migration index along the *y*-axis (yFMI), a measure for how directed each track is with respect to the gradient, as well as the Center of Mass for the total cell population.

Quantification of chemotaxis outside the observation chamber (Figs. 3d, 4c) was done by cropping the images to an area of 2.29 × 0.78 mm immediately above the observation area (Supplementary Fig. 6C). Then a line was drawn at 2/3 of the area height, and the percentage of cells crossing that line over the course of the 180 min movie was determined.

**RNA immunoprecipitation**. Cells in chemotactic streams (from ten pooled 15 cm culture dishes) were lysed in 500 µl lysis buffer (25 mM HEPES pH 7.4, 150 mM NaCl, 3 mM EGTA, 3 mM EDTA, 1% Triton-X 100, 40 U RNaseOut (Thermo Fisher Scientific, Waltham, USA) and Complete Protease Inhibitor Cocktail (Roche)). The lysate was cleared by centrifugation at 10,000 × g for 8 min. Supernatant was incubated for 1 h with 150 µl Protein A-Sepharose beads which had been coupled for 1 h with mouse anti-GFP antibody (1:10 dilution, Roche, Switzerland) for the GFP-Puf118-RBD pull down, and with mouse anti-myc antibody (1:10 dilution, Roche, Switzerland) for the myc-TalA-Puf118 pull down. For the myc-TalA-Puf118 pull down, mouse IgG (1:10 dilution; Abcam, Cambridge, UK) was used as a control. After incubation, beads were washed six times with lysis buffer. For western blotting, 2% of the total lysate and 5% of the beads were separated by SDS-PAGE, and blotted with anti-myc or anti-GFP antibodies. RNA

was extracted from the remainder of the beads with TRIzol (Thermo Fisher, Waltham, USA) according to manufacturer's instructions. The RNA pellet was resuspended in 8 µl of H$_2$O, DNAse treated for 30 min at 37 °C (1 U/10 µl DNase I; Thermo Fisher Scientific, Waltham, USA) and then DNAse was inactivated by addition of 1 mM EDTA and incubation at 65 °C for 15 min. Then the RNA was incubated with 20 µM poly-T primer and dNTPs for 15 min at 65 °C, followed by an annealing step on ice for 1 min. Next, reverse transcription was performed with SuperscriptIII reverse transcriptase (Invitrogen) according to manufacturer's instructions. To detect the individual mRNAs, primer pairs specific for each of the target mRNAs that yielded a product of ~300 bp were used for PCR (see Supplementary Table 2; Phusion, Thermo Fisher Scientific, Waltham, USA). All uncropped Agarose gels from this study can be found in Supplementary Fig. 10.

**Determination of mRNA and protein levels**. Total levels of chemotaxis pathway mRNAs were determined on lysates of wild-type cells expressing + or -PBE variants with or without coexpression of the TalA-Puf118 construct in lysis buffer as above. Total mRNA was extracted using TRIzol (Thermo Fisher Scientific, Waltham, USA) according to manufacturer's instructions and amounts were measured with a Nanodrop 2000 spectrophotometer (Thermo Fisher Scientific, Waltham, USA). One microgram total mRNA was DNAse treated for 30 min at 37 °C (1 U/10 µl DNase I; Thermo Fisher Scientific, Waltham, USA) and reverse transcription was done as above. Then we performed RT-qPCR using 2.5 µl of 1:20 diluted cDNA with iTaq Universal SYBR Green Supermix (Bio-Rad, Hercules, USA) according to manufacturer's instructions. RT-qPCRs were performed with a Lightcycler 480 II (Roche, Switzerland) using the following program: Pre-incubation: 95 °C for 5 min with a ramp rate of 4.8 (°C/s); amplification: 45 cycles at 95 °C for 15 s with a ramp rate of 2.4 °C/s, 60 °C for 15 s with a ramp rate of 2.4 °C/s, 72 °C for 15 s with a ramp rate of 2.4 °C/s; melting curve: 95 °C for 5 s with a ramp rate of 4.8 °C/s; 65 °C for 1 min with a ramp rate of 2.2 °C/s; 97 °C with a ramp rate of 0.11 °C/s; 5 Acquisitions/C. All RT-qPCRs were done in technical triplicates and included a negative control (no RT) for each of the tested RNAs. $C_q$ values were determined for each sample, Mean and standard deviation of technical triplicates were taken and normalized to the $C_q$ value obtained for ScdA in each respective sample. All negative controls gave no $C_q$ value and are therefore not shown. To analyze the protein levels of chemotaxis pathway components we used cell lysates as above. Protein content was determined by Quick Start Bradford Protein Assay (Bio-Rad, Hercules, USA) according to manufacturer's recommendation, and the lysates of wild-type +PBE, wild-type -PBE and TalA-Puf118 +PBE were adjusted relative to each other for every protein. Samples were then analyzed by SDS-PAGE and blotted against GFP as above. An anti-IQGAP1 antibody was used to ensure equal loading. All uncropped Western blots from this study can be found in Supplementary Fig. 10.

**Database searches**. To identify *D. discoideum* homologs of yeast Puf4, BLAST function on Dictybase (http://dictybase.org/tools/blast) was used to search the genome with the RNA-binding domain of yeast Puf4 as a query. The in silico screen for PBEs in *D. discoideum*, was performed by manual sequence analysis of all genes annotated with the function "Chemotaxis" (97 genes), "Metabolism" (58 genes), or "Mitochondria" (93 genes). Consensus recognition sites for Puf proteins were based on known yeast PBE sites (Supplementary Fig. 2A, ref. [20]) and one variation from the exact consensus site for Puf3, Puf4, or Puf5 binding was allowed. Since the exact length of the 3′-UTR for each gene is unknown, we only considered PBEs within 330 bp after the STOP codon, assuming that the average length of a 3′-UTR was comparable to that in yeast[35]. To search for potential zip codes in the 3′-UTRs of chemotaxis mRNAs, we manually searched the 500 base pairs following the STOP codon for the occurrence of the motifs 5′-Pyrimidine-CACCC-3′[18] and 5′-CGGAC-3′[17].

**Data availability**. All the data supporting the findings of this study are included in the Figures or Supplementary Note 1, or can be obtained directly from the authors upon request.

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

## Acknowledgements

M.H. was supported by a Swiss National Science Foundation Fellowship (#PBEZP3_142925), and work in the Nelson Laboratory was supported by the NIH (R35GM118064). We thank Dr. A. Kortholt for providing the *pla2/sgcA*-null cells, Dictybase for plasmid constructs, Dr. M. Koeberlin for help with RT-qPCR and members of the Nelson lab, particularly Dr. A. Mueller, for support and fruitful discussions.

## Author contributions

M.H. and W.J.N. designed the experiments, M.H. performed the experiments, and M.H. and W.J.N. interpreted the results and wrote the manuscript.

## Additional information

**Competing interests:** The authors declare no competing financial interests.

