## [Peer Review File · Nature Communications]

Reviewers' Comments:

Reviewer #1:

Remarks to the Author:

The manuscript by Hotz and Nelson describes a very interesting study on the possible role of mRNAs localization by Tuf118 in chemotaxis. This study is potentially very important and changing the field. However differences in protein localization between front and rear are small, the data are not analyzed and presented correctly, several controls are missing, and the explanation of the effects is not convincing.

Major points.

1. The data are presented as the ratio of intensities in the front and rear on a linear scale. This is not correct. The data should be analyzed and presented as the logarithm of the F/R ratio, in a way -and for the same reason- as is done in for instance proteomic data. The reason is that if the intensity and SD of a GFP-tagged protein have a normal distribution in the front and in the rear, the mean and SD of the F/R ratio is not a normal distribution. It not only leads to mistakes in calculating the SD (they are asymmetric on a linear scale as can be seen in all the figures), but it also leads to the wrong calculation of the mean. This can be demonstrated with an example. Assume the intensities at the front and rear are both 15 ± 5 with normal distributions. An intensity of 20 is equally likely as an intensity of 10, and these intensities are equally likely in the front and rear. Thus a F/R ratio = $20/10 = 2$ is equally likely as F/R ratio = $10/20 = 0.5$. The mean of these two ratios is not the expected 1.0 but 1.25. And also the inverse mean R/F ratio = 1.25. It can be calculated that with $F = 15 \pm 5$ and $R = 15 \pm 5$ the expected mean of F/R = 1.16. This anomaly disappears when analyzing $\log(F/R)$, because the mean of $\log(2)$ and $\log(0.5)$ equals $\log(1)$, and the SD will be symmetric.

The authors should convert all primary F/R data to $\log(F/R)$ value and then take the mean and SD, and present the data on a $\log(F/R)$ scale.

2. The intensities of the proteins is determined on fixed specimen only. This was probably done because RNA- FISH needs fixed material. However, experiments on protein localization are also possible and should be performed on life cells (as was apparently done for Puf118-GFP in figure S3). See also next point.

3. Many images show in the rear of the cell vacuole-like structures devoid of protein-GFP, which are not or much less detectable in the organelle free area of the protrusions in the front of the cell. It is well possible that pixels in the rear of the cell have contain cytosol that in the front of the cell. Therefore experiments on life cells without fixation are required. In addition, the experiments described above for GFP-tagged protein localization in life cells should include an internal control with cytosolic RFP. The data should be calculated and presented first as GFP/RFP ratio, and then this ratio is determined in the front and rear of the cell and presented as the log value thus $\log[(\text{GFP/RFP front})/(\text{GFP/RFP rear})]$.

4. The data are explained with the assumption that proteins are synthesized at Tub118 in the front of the cell and remain active in the front of the cell. However chemotaxis has a very short time constant; cells can change direction within a few seconds upon stimulation with a new gradient, but these proteins have a life-time of several hours. So how do the protein stay in the front of the cell if the front changes position rapidly. This should also be analyzed experimentally by observing the localization of e.g. GFP-PLA2 in life chemotaxis cells and let the cells make a new direction.

The experiment with (-)PBE and (+)PBE mRNA for chemotaxis proteins is intriguing. The observation of different localization of these mRNAs is understandable, as (+)PBE binds to TUB118 that is localized in the front. But the logic for different protein localization is not easily understood. The proteins synthesized from (-)PBE mRNA are not different from proteins synthesized from (+)PBE. So how can they localize differently?

(+)PBE-GFP localizes to the front, the RNA localizes to the front, but the GFP protein not. This suggests that the place of GFP synthesis is not relevant for its later localization, as expected. However, this apparently does not hold for the chemotaxis proteins that remain localized at the position of Tub118 where they were synthesized.

A possible explanation of the observations (if they hold with the more stringent analysis and controls mentioned above) is the hypothesis that Tuf118 with its subset of mRNAs binds to F-actin, synthesizes the chemotaxis proteins in the front and that these proteins remain bound to Tuf118. When F-actin re-localizes during chemotaxis, Tuf118 with its mRNAs and associated proteins follows this re-localization.

Do Tub118-synthesized chemotaxis proteins bind to the Tub118 protein? Can they be identified in Tub118 IPs?

5. The TalA experiments are tantalizing but also intriguing. The localization of TalA-Tub118 to the rear is associated with rearward localization of chemotaxis proteins and slightly defective chemotaxis. However, localization of TalA-Tub118 is strange. TalA itself localizes to myosinII and is detectible as a narrow layer in the cortex just under the plasma membrane. The localization of TalA-Puf118 is very different and appears as an oval clot in the cytoplasm. A control with the localization of myc-tagged TalA is missing, to be compared with the localization of TalA -GFP (cortex) and myc-tagged TalA-Tub118. The experiments on the chemotactic effect of myc-tagged TalA-Tub118 lack the control of chemotaxis of cells expressing myc-tagged TalA without fusion to Tub118.

Minor comments

1. The purple bar in several figures at F/R = 1 is misleading. The bar has a width suggesting significance (SD of control). However, the width has no information and is thus misleading. Replace the bar by a dotted line.

2. Figure S3 showing kymograph of Puf118-GFP should also present images of the cell, to provide a reference for the reader with the images in the main manuscript.

Reviewer #2:

Remarks to the Author:

Summary and critiques:

The manuscript "Pumilio-dependent localization of mRNAs at the cell front co-ordinates multiple pathways required for chemotaxis" by Hotz and Nelson investigates the role of the PUF/Pumilio family member Puf118 in localization of specific mRNAs and proteins during chemotaxis in *Dictyostelium discoideum* (Dd).

Four pathways participate in chemotaxis in Dd, and the authors observed that protein components of each pathway are localized to the "front" of migrating cells, as do the mRNAs encoding these proteins. Likewise, one of multiple actin mRNAs and its encoded protein are localized to the front. This is reminiscent of the Zip-code mediated localization of Actin mRNA reported by Rob Singer's group, and links to a growing array of examples of active mRNA localization in the control of subcellular localization of protein expression.

The authors report a role for the Dd Puf118 in coordination of localization of several mRNAs encoding chemotaxis pathway components. Puf118 is itself localized to the front and actin polymerization is necessary for this effect. The RNA binding specificity of Dd Pufs is unknown, and the authors use RNA binding consensus sequences from several *Saccharomyces* Pufs as search motifs to find that sequences resembling yeast Puf binding sites are prevalent in Dd chemotaxis genes. These are presumed to be PBEs for Puf118, which is reported here to be the only cytoplasmic Dd Puf. This is useful for hypothesis generation but the authors should acknowledge that the specificity of Dd Puf118 could be quite different from the yeast Pufs. Still, they find that the mRNA/protein localization is dependent on certain PBE elements in the 3'UTR of the affected

mRNAs.

Evidence is presented to suggest that Puf118 interacts with these mRNAs, but the specific role of the PBEs in this interaction is not addressed. Interpretation of these results is hindered by missing controls (see below). It would be useful to show that mutation of the PBEs blocks Puf118-mRNA co-immunoprecipitation. This assay or in vitro binding assays are necessary to prove that the presumed PBEs are in fact Puf118 binding sites. Mutations of the presumed PBEs does reduce mRNA and protein localization during chemotaxis, and mutation of the PBE in Pla2 mRNA reduced migration in a gradient of cAMP, supporting the functional role of the presumed PBE in the chemotaxis.

Ectopic localization of the Puf118 RNA binding domain to the rear of the cells causes mislocalization of the target mRNAs and proteins, and disrupts migration directionality and chemotaxis.

This work presents a novel role of Dd Puf118 in chemotaxis. In broader sense, PUF proteins themselves have been linked to RNA localization in yeast, including asymmetric cell division and mitochondrial localization. In addition, mammalian PUFs have been reported to participate in actin dependent mRNA localization in neurons. Those findings do not detract from the importance of this work, and those precedents should be cited. The findings of Hotz and Nelson also match the post-transcriptional operon hypothesis put forth by Jack Keene (PMID: 17572691), and thus it would be appropriate to cite that work as well. Overall the citations in the manuscript are sparse.

The manuscript is well-written, the data are clearly presented, and in most cases proper statistical analysis is presented.

Beyond the role of Puf118, the dependence on actin polymerization and certain presumed PBEs, the mechanism of localization of these transcripts remains unexplored.

Major Critique:

The following issues must be addressed:

A major issue in this work involves the detection and attempted quantitation of RNA. RT-PCR is very sensitive to false-positive signal due to contamination by genomic and/or plasmid DNA. The authors use Trizol for RNA purification, which is plagued by DNA contamination. No DNase treatment is used on the purified RNA to reduce/eliminate contamination. Moreover, the universal, necessary control for any RT-PCR experiment is a no-RT control for each assay. As such, it is impossible to discern if the reported signals are amplified from RNA or contaminating DNA. Further, the RT-PCR assays are not adequately described in the methods.

The authors conclude the Puf118 has no effect on RNA levels but this conclusion is not supported. As above, they did not include negative -RT control. Second, they use endpoint RT-PCR (again, PCR conditions and cycle numbers are not reported). It is not possible to measure mRNA levels from a single endpoint RT PCR assay. qRT-PCR, Northern blot, or other methods are necessary to discern whether Puf118 affects mRNA levels. Moreover, given the widely observed effects of Pufs on reducing RNA stability, a transcriptional shutoff assay would be advisable. Moreover, the endpoint RT-PCR assays appear to be from 1 replication, as no stats or error bars are reported.

Reviewer #3:

Remarks to the Author:

Manuscript background information

In this important manuscript, Hotz and Nelson investigated the mechanisms underlying cAMP dependent chemotaxis of *Dictyostelium discoideum* (Dicty). They sought to determine a common factor that linked the four different pathways (PI3-kinases (PI3K), TorC, the phospholipase Pla2, and the guanylyl cyclases SgcA and GcA) together. This coregulation would allow them to work synergistically downstream of RasC signaling, allowing Dicty to have persistent directed migration in weak cAMP gradients. Upon GFP-tagging and RNA FISH for the major proteins and mRNAs involved in each pathway, the authors noticed that chemotaxis pathways have both localized proteins and mRNAs that resemble the polarization of phalloidin stained F-actin / Act1-GFP.

To investigate the mechanism regulating this RNA localization and localized translation they

searched the 3' UTRs of these genes to see if they contained common RNA localization motifs. The authors found that the PUF localization sequences homologous to yeast were present in the different UTRs in question and were highly enriched in annotated chemotaxis genes across the Dicty genome. To determine which of the Puf genes might be responsible for this localization they tested the protein localization of each of the 5 Puf genes and found that only a previously uncharacterized protein they termed Puf118 localized to the cytoplasm. As this gene is homologous to yeast Puf4 they proceeded under the hypothesis that the two proteins may share a consensus sequence.

To validate the importance of Puf118, a number of experiments were performed. First, the localized RNAs that were previously tested were shown to immunoprecipitate with GFP tagged Puf118. While the 3' UTRs containing PBEs were sufficient to localize the RNA, they did not confer localization upon their GFP reporter, indicating that a separate mechanism retains the chemotaxis proteins independent of the RNAs. Deletion of the binding element within the 3' UTRs of the reporter genes caused a loss of polarization in a KO background. The authors then showed that by mislocalizing Puf118 to the rear of the cell (by fusing it to Tala), the opposite RNA polarity could be achieved.

Comments for the authors

This is an excellent manuscript that provides a unifying theory and mechanism for chemotaxis in Dicty, a model organism that is useful for the study of cell migration. The authors provide evidence that all 4 chemotaxis pathways are regulated by the same, previously uncharacterized RNA binding protein (Puf118) and validate their conclusions in multiple ways. Their findings will be a valuable addition to the field of mRNA mediated polarized migration. In particular it adds a mechanistic component that is an important rationale for localized translation to establish polarity in a chemotactic gradient. Back in the last century, Condeelis and I debated whether Dicty could use mRNA localization to mediate chemotaxis as with fibroblasts and he argued me out of it, claiming that Dicty is to a fibroblast as a motorboat is to a battleship, and hence would be refractory to localized protein synthesis, which would be too slow. It's great to prove him wrong! This should be published right away.

A few minor issues resolvable by some text tweaking: the manuscript could use a bit more clarification and detail to better explain how the authors arrived at Puf118 as their protein of interest. There seem to be some logical leaps involving yeast, and that may be the sum of it, but if there is some more directed investigation, that would be informative in the supplement. For instance, the authors find that Puf118 is most similar to yeast Puf4 but the methods by which they made this comparison are vague. Was this based on the nucleotide or amino acid sequence? It would also be helpful if the authors provided more details on how they performed their search for zipcodes / binding motifs? Was it a manual search or a brute force approach? How many localization sequences were tested (apart from the zipcodes and PBEs)? The PUF motif has rather low complexity, it would be expected to be common throughout the genome. The genome wide search that was performed in supplementary figure 1 compares genes related to mitochondria, metabolism and chemotaxis. Were these the three most enriched categories? Is RNA localization and chemotaxis the major function of Puf118? It may be helpful to add the sequence homologies for each of the five Dicty Puf genes into figure S2. Figure S2 also begs the question about why the remaining four Dicty Puf genes are nuclear (as mentioned in the text) and why only two of the four remaining genes were shown in the figure. How similar are these other genes compared to Puf118 and can these differences hint at possible unique functions? Do they also have homologies to Yeast Puf proteins? The paper relies on an initial assumption that yeast Puf4 and Puf118 have the same consensus motif but this was not demonstrated directly.

The connection between shallow/steep gradient in terms of normal Dicty migration is not totally clear. What biological function does the shallow vs steep gradient represent? The authors show that RasC (the upstream regulator of the 4 signaling pathways) protein and RNA does not localize

in the shallow cAMP gradient. Was this also tested in a steep cAMP gradient also? Could the localization in a steep gradient be Puf118 independent or does this occur through a separate mechanism? This is important to show because Dicty is able to rapidly change its localization direction. The mechanism that the authors propose with Puf118 seems more suitable for long term / persistent migration. There is some similarity to fibroblasts, a non-polarized version lacking ZBP1 can be induced to polarize in a strong chemotactic gradient (Lapidus et al., ZBP1 enhances cell polarity and reduces chemotaxis, PMC4956933). The referencing is a bit sketchy in covering the field. I suggest that they consider referencing the following publication that relates mRNA localization to motility by real time image analysis: "An unbiased analysis method to quantify mRNA localization reveals its correlation with cell motility" Park, et al. PMID:22832165.

Finally, it may be worthwhile to relate these findings back to the larger picture. The model that the authors provide is helpful but do the authors expect this to be conserved in other organisms? One example that comes to mind is neutrophil chemotaxis, is there any evidence for a similar mechanism in human cells?

The work shows quite clearly that the role of mRNA localization is to bring together many components of the motility mechanism. This is an important concept that we have argued functions with the actin based motility of lamellipods, in response to criticism that synthesis of actin is insufficient to explain protrusion. We posited that it's not only actin but all the many migration related proteins that together are responsible for directed migration. This work gives experimental credence for this concept. I think it should be emphasized in the discussion.

In conclusion this manuscript will have broad appeal to investigators many fields including researchers studying Dicty, chemotaxis, pumilio binding proteins and RNA localization.

Sorry I missed your talk at Einstein. This makes up for it!
Rob Singer

REBUTTAL

The reviewers' critiques are copied *verbatim* in black font, and our responses are in red font

Reviewers' comments:

Reviewer #1 (Remarks to the Author):

The manuscript by Hotz and Nelson describes a very interesting study on the possible role of mRNAs localization by Tuf118 in chemotaxis. This study is potentially very important and changing the field. However differences in protein localization between front and rear are small, the data are not analyzed and presented correctly, several controls are missing, and the explanation of the effects is not convincing.

Major points.

1. The data are presented as the ratio of intensities in the front and rear on a linear scale. This is not correct. The data should be analyzed and presented as the logarithm of the F/R ratio, in a way -and for the same reason- as is done in for instance proteomic data. The reason is that if the intensity and SD of a GFP-tagged protein have a normal distribution in the front and in the rear, the mean and SD of the F/R ratio is not a normal distribution. It not only leads to mistakes in calculating the SD (they are asymmetric on a linear scale as can be seen in all the figures), but it also leads to the wrong calculation of the mean. This can be demonstrated with an example. Assume the intensities at the front and rear are both 15 ± 5 with normal distributions. An intensity of 20 is equally likely as an intensity of 10, and these intensities are equally likely in the front and rear. Thus a F/R ratio = $20/10 = 2$ is equally likely as F/R ratio = $10/20 = 0.5$. The mean of these two ratios is not the expected 1.0 but 1.25. And also the inverse mean R/F ratio = 1.25. It can be calculated that with $F = 15 \pm 5$ and $R = 15 \pm 5$ the expected mean of F/R = 1.16. This anomaly disappears when analyzing $\log(F/R)$, because the mean of $\log(2)$ and $\log(0.5)$ equals $\log(1)$, and the SD will be symmetric. The authors should convert all primary F/R data to $\log(F/R)$ value and then take the mean and SD, and present the data on a $\log(F/R)$ scale.

We thank the reviewer for this comment. We fully agree that showing $\log(F/R)$ is correct to obtain a measurement of the degree of asymmetric distribution of proteins and mRNAs. We have, therefore, changed all F/R data to $\log(F/R)$ in the figures and text. However, we do not present the data on a log scale, since negative values cannot be shown on a log scale. In all cases, the data remain statistically significant.

2. The intensities of the proteins is determined on fixed specimen only. This was probably done because RNA- FISH needs fixed material. However, experiments on protein localization are also possible and should be performed on live cells (as was apparently done for Puf118-GFP in figure S3). See also next point.

We performed live cell microscopy on cells expressing Pla2-GFP or Lst8-GFP, and include the same analysis as we had done for Puf118-GFP (kymograph and time-lapse asymmetry quantification, Fig. S1). The results from live-cell imaging are comparable to those from fixed cell images.

3. Many images show in the rear of the cell vacuole-like structures devoid of protein-

GFP, which are not or much less detectable in the organelle free area of the protrusions in the front of the cell. It is well possible that pixels in the rear of the cell have contain cytosol that in the front of the cell. Therefore experiments on life cells without fixation are required. In addition, the experiments described above for GFP-tagged protein localization in life cells should include an internal control with cytosolic RFP. The data should be calculated and presented first as GFP/RFP ratio, and then this ratio is determined in the front and rear of the cell and presented as the log value thus $\log[(\text{GFP/RFP front})/(\text{GFP/RFP rear})]$.

Overall, we did not observe an asymmetric distribution of the vacuole in chemotaxing cells. In order to control for potential geometric effects leading to an overestimation of asymmetry (see lines 42-42), we used: 1) A cytosolic GFP reporter construct (Figure 1A), showing that the distributions of both GFP mRNA in live and fixed cells were symmetric (i.e. diffuse throughout the cytoplasm, with no polarized accumulation at the cell front); 2) RasC mRNA and protein (Fig. 1A), showing that the distributions were symmetric and uniformly diffuse (confirming earlier reports); and 3) All of the -PBE mutant constructs (Figure 3A and B), showing that their distributions were symmetric and uniformly diffuse. Thus the front/rear polarization observed for chemotaxis mRNAs and proteins is not a geometric effect, but indeed a regulated process that depends on the PBE.

4. The data are explained with the assumption that proteins are synthesized at Tub118 in the front of the cell and remain active in the front of the cell. However chemotaxis has a very short time constant; cells can change direction within a few seconds upon stimulation with a new gradient, but these proteins have a life-time of several hours. So how do the protein stay in the front of the cell if the front changes position rapidly. This should also be analyzed experimentally by observing the localization of e.g. GFP-PLA2 in life chemotaxis cells and let the cells make a new direction.

We wanted to examine cells responding physiologically to a natural chemotaxis gradient. Thus, we performed all localization studies in natural chemotaxis gradients, i.e. gradients formed by the cells as a population, rather than from an artificial external point source of a very high concentration of cAMP. Since the gradients generated by cells are natural and many co-existing cAMP sources create a dynamic environment, the cells are continuously exposed to subtle changes in gradient direction and, hence reorientation of the cell front; this can be seen in the still images in Fig. 2B. We observed that the polarized distributions of mRNA and proteins at the cell front were indeed dynamic and occurred in a Puf118-dependent fashion, indicating that this regulatory mechanism is indeed within the time-scale of dynamic cell migration that occurs in a natural chemotaxis gradient.

The experiment with (-)PBE and (+)PBE mRNA for chemotaxis proteins is intriguing. The observation of different localization of these mRNAs is understandable, as (+)PBE binds to TUB118 that is localized in the front. But the logic for different protein localization is not easily understood. The proteins synthesized from (-)PBE mRNA are not different from proteins synthesized from (+)PBE. So how can they localize differently? (+)PBE-GFP localizes to the front, the RNA localizes to the front, but the GFP protein not. This suggests that the place of GFP synthesis is not relevant for its later localization, as expected. However, this apparently does not hold for the chemotaxis proteins that remain localized at the position of Tub118 where they were synthesized.

The reviewer raises an intriguing point that we also identified during the work. We postulate that proteins are synthesized in the cell front where their mRNAs bind to Puf118. This is consistent with the result that perturbing mRNA polarization by mutation of the PBE leads to symmetric distribution of both mRNA and protein. Why proteins remain in the front (as opposed to GFP protein which rapidly diffuses – Fig. 3B) remains unclear, as it is for many other locally synthesized proteins in other systems. Our work provides strong evidence that mRNAs required for chemotaxis are localized at the cell front by Puf118 and that mislocalization of those mRNAs blocks chemotaxis; this conclusion is in itself novel, and we feel that further analysis of mechanisms involved in retaining proteins at the cell front is beyond the scope of the present work.

A possible explanation of the observations (if they hold with the more stringent analysis and controls mentioned above) is the hypothesis that Tuf118 with its subset of mRNAs binds to F-actin, synthesizes the chemotaxis proteins in the front and that these proteins remain bound to Tuf118. When F-actin re-localizes during chemotaxis, Tuf118 with its mRNAs and associated proteins follows this re-localization. Do Tub118-synthesized chemotaxis proteins bind to the Tub118 protein? Can they be identified in Tub118 IPs?

As we noted in response to the previous comment, we have been thinking about this problem too and also considered the possibility of a role for actin (see the model in Fig S9). However, there is no indication that any of these proteins bind actin, and none have canonical actin-binding domains based on amino acid sequence. Also, to our knowledge RNA-binding proteins are usually absent from the mRNAs once the translation machinery is loaded (Wu et al., 2015; PMID: 26140598), and thus the idea that proteins remain associated to the RNA-binding protein that bound the mRNA seems unlikely. We have added this point to the text (lines 99-101). As we noted above, our conclusion on the requirement for front localization of Puf118-dependent chemotaxis mRNAs is in itself novel, and we feel that further analysis of mechanisms retaining proteins is beyond the scope of the present work.

5. The TalA experiments are tantalizing but also intriguing. The localization of TalA-Tub118 to the rear is associated with rearward localization of chemotaxis proteins and slightly defective chemotaxis. However, localization of TalA-Tub118 is strange. TalA itself localizes to myosinII and is detectible as a narrow layer in the cortex just under the plasma membrane. The localization of TalA-Puf118 is very different and appears as an oval clot in the cytoplasm. A control with the localization of myc-tagged TalA is missing, to be compared with the localization of TalA –GFP (cortex) and myc-tagged TalA-Tub118. The experiments on the chemotactic effect of myc-tagged TalA-Tub118 lack the control of chemotaxis of cells expressing myc-tagged TalA without fusion to Tub118.

We agree that the localization of the TalA-Puf118 construct is not identical to GFP-tagged TalA as described in Tsuijoka et al. (2012), and we have accordingly added a comment in the text (line 124-126). Nevertheless, for the purpose of our experiments this construct is fully sufficient to test our hypothesis, since TalA-Puf118 localizes consistently to the rear of migrating cells and serves as a reliable tool to mislocalize the RNA-binding domain of Puf118; significantly, mRNA mislocalization inhibited chemotaxis, thus supporting our analysis that Puf118-dependent localization of these mRNAs to the cell front is required for directed cell migration in a chemotactic gradient.

Minor comments

1. The purple bar in several figures at $F/R = 1$ is misleading. The bar has a width

suggesting significance (SD of control). However, the width has no information and is thus misleading. Replace the bar by a dotted line.

We used a grey bar to represent the range of 'symmetric' mRNA or protein localization (0.9 to 1.1 on the y-axis); a dotted line is difficult to see with the many other lines and data points. We have stated in the figure legend and in the new figures the range is shown from $\log(0.9)$ to $\log(1.1)$ (lines 264, 279, 291: "The grey area...").

2. Figure S3 showing kymograph of Puf118-GFP should also present images of the cell, to provide a reference for the reader with the images in the main manuscript.

This has been added to the figure.

Reviewer #2 (Remarks to the Author):

Summary and critiques:

The manuscript "Pumilio-dependent localization of mRNAs at the cell front co-ordinates multiple pathways required for chemotaxis" by Hotz and Nelson investigates the role of the PUF/Pumilio family member Puf118 in localization of specific mRNAs and proteins during chemotaxis in *Dictyostelium discoideum* (Dd).

Four pathways participate in chemotaxis in Dd, and the authors observed that protein components of each pathway are localized to the "front" of migrating cells, as do the mRNAs encoding these proteins. Likewise, one of multiple actin mRNAs and its encoded protein are localized to the front. This is reminiscent of the Zip-code mediated localization of Actin mRNA reported by Rob Singer's group, and links to a growing array of examples of active mRNA localization in the control of subcellular localization of protein expression.

The authors report a role for the Dd Puf118 in coordination of localization of several mRNAs encoding chemotaxis pathway components. Puf118 is itself localized to the front and actin polymerization is necessary for this effect. The RNA binding specificity of Dd Pufs is unknown, and the authors use RNA binding consensus sequences from several *Saccharomyces* Pufs as search motifs to find that sequences resembling yeast Puf binding sites are prevalent in Dd chemotaxis genes. These are presumed to be PBEs for Puf118, which is reported here to be the only cytoplasmic Dd Puf. This is useful for hypothesis generation but the authors should acknowledge that the specificity of Dd Puf118 could be quite different from the yeast Pufs.

This is a good point, and we have acknowledged that the specificity of Dd Puf118 could be different from the yeast Pufs in the text (line 59: "Assuming..."). However, we also note in the text (line 62-65) that the RBD sequence of Puf118 is very similar to that of Puf4 (46% identical and 64% similar residues), and the residues known to be responsible for mRNA target recognition are nearly identical (90% identical and 96% similar residues; Fig. S3C).

Still, they find that the mRNA/protein localization is dependent on certain PBE elements in the 3'UTR of the affected mRNAs.

Evidence is presented to suggest that Puf118 interacts with these mRNAs, but the specific role of the PBEs in this interaction is not addressed. Interpretation of these results is hindered by missing controls (see below). It would be useful to show that

mutation of the PBEs blocks Puf118-mRNA co-immunoprecipitation. This assay or in vitro binding assays are necessary to prove that the presumed PBEs are in fact Puf118 binding sites.

We understand the point that the reviewer makes. However, we decided that a more direct and stringent test, and one that we could assess functionally in the context of chemotaxis, was to mis-localize chemotactic mRNAs to the rear of the cell using Tala-Puf118. This indeed occurred, and, importantly, chemotactic mRNA (and protein) mislocalization depended on the presence of the PBE (Fig. S7). We then showed that these cells were completely defective in chemotaxis. We believe that this critical experiment is strong evidence that the interaction between Puf118 and the mRNAs is PBE-dependent, and that it is required for chemotaxis.

Mutations of the presumed PBEs does reduce mRNA and protein localization during chemotaxis, and mutation of the PBE in Pla2 mRNA reduced migration in a gradient of cAMP, supporting the functional role of the presumed PBE in the chemotaxis. Ectopic localization of the Puf118 RNA binding domain to the rear of the cells causes mislocalization of the target mRNAs and proteins, and disrupts migration directionality and chemotaxis.

This work presents a novel role of Dd Puf118 in chemotaxis. In broader sense, PUF proteins themselves have been linked to RNA localization in yeast, including asymmetric cell division and mitochondrial localization. In addition, mammalian PUFs have been reported to participate in actin dependent mRNA localization in neurons. Those findings do not detract from the importance of this work, and those precedents should be cited. The findings of Hotz and Nelson also match the post-transcriptional operon hypothesis put forth by Jack Keene (PMID: 17572691), and thus it would be appropriate to cite that work as well. Overall the citations in the manuscript are sparse.

Thank you for identifying the Keen reference which we have added the text. We have also added more citations to other studies.

The manuscript is well-written, the data are clearly presented, and in most cases proper statistical analysis is presented.

Beyond the role of Puf118, the dependence on actin polymerization and certain presumed PBEs, the mechanism of localization of these transcripts remains unexplored.

Major Critique:

The following issues must be addressed:

A major issue in this work involves the detection and attempted quantitation of RNA. RT-PCR is very sensitive to false-positive signal due to contamination by genomic and/or plasmid DNA. The authors use Trizol for RNA purification, which is plagued by DNA contamination. No DNase treatment is used on the purified RNA to reduce/eliminate contamination. Moreover, the universal, necessary control for any RT-PCR experiment is a no-RT control for each assay. As such, it is impossible to discern if the reported signals are amplified from RNA or contaminating DNA. Further, the RT-PCR assays are not adequately described in the methods.

We had performed DNase treatment for all our RT-PCRs in our original experiments, but this step was accidentally omitted from the text. It is now described in detail in Materials and Methods.

The authors conclude the Puf118 has no effect on RNA levels but this conclusion is not supported. As above, they did not include negative –RT control. Second, they use endpoint RT-PCR (again, PCR conditions and cycle numbers are not reported). It is not possible to measure mRNA levels from a single endpoint RT PCR assay. qRT-PCR, Northern blot, or other methods are necessary to discern whether Puf118 affects mRNA levels. Moreover, given the widely observed effects of Pufs on reducing RNA stability, a transcriptional shutoff assay would be advisable. Moreover, the endpoint RT-PCR assays appear to be from 1 replication, as no stats or error bars are reported.

This is a fair point, and we performed qRT-PCR. We have replaced the RT-PCR experiment on the total mRNA levels with qRT-PCR to reliably determine the mRNA levels in +PBE and -PBE mutants, as well as in TalA-Puf118 expressing cells. These experiments included a negative-RT control and showed that the levels are unchanged between the various mutants. This confirms our original observation.

Reviewer #3 (Remarks to the Author):

Manuscript background information

In this important manuscript, Hotz and Nelson investigated the mechanisms underlying cAMP dependent chemotaxis of *Dictyostelium discoideum* (Dicty). They sought to determine a common factor that linked the four different pathways (PI3-kinases (PI3K), TorC, the phospholipase Pla2, and the guanylyl cyclases SgcA and GcA) together. This coregulation would allow them to work synergistically downstream of RasC signaling, allowing Dicty to have persistent directed migration in weak cAMP gradients. Upon GFP-tagging and RNA FISH for the major proteins and mRNAs involved in each pathway, the authors noticed that chemotaxis pathways have both localized proteins and mRNAs that resemble the polarization of phalloidin stained F-actin / Act1-GFP.

To investigate the mechanism regulating this RNA localization and localized translation they searched the 3' UTRs of these genes to see if they contained common RNA localization motifs. The authors found that the PUF localization sequences homologous to yeast were present in the different UTRs in question and were highly enriched in annotated chemotaxis genes across the Dicty genome. To determine which of the Puf genes might be responsible for this localization they tested the protein localization of each of the 5 Puf genes and found that only a previously uncharacterized protein they termed Puf118 localized to the cytoplasm. As this gene is homologous to yeast Puf4 they proceeded under the hypothesis that the two proteins may share a consensus sequence.

To validate the importance of Puf118, a number of experiments were performed. First, the localized RNAs that were previously tested were shown to immunoprecipitate with GFP tagged Puf118. While the 3' UTRs containing PBEs were sufficient to localize the RNA, they did not confer localization upon their GFP reporter, indicating that a separate mechanism retains the chemotaxis proteins independent of the RNAs. Deletion of the binding element within the 3' UTRs of the of the reporter genes caused a loss of polarization in a KO background. The authors then showed that by mislocalizing Puf118 to the rear of the cell (by fusing it to TalA), the opposite RNA polarity could be achieved.

Comments for the authors

This is an excellent manuscript that provides a unifying theory and mechanism for

chemotaxis in Dicty, a model organism that is useful for the study of cell migration. The authors provide evidence that all 4 chemotaxis pathways are regulated by the same, previously uncharacterized RNA binding protein (Puf118) and validate their conclusions in multiple ways. Their findings will be a valuable addition to the field of mRNA mediated polarized migration. In particular it adds a mechanistic component that is an important rationale for localized translation to establish polarity in a chemotactic gradient. Back in the last century, Condeelis and I debated whether Dicty could use mRNA localization to mediate chemotaxis as with fibroblasts and he argued me out of it, claiming that Dicty is to a fibroblast as a motorboat is to a battleship, and hence would be refractory to localized protein synthesis, which would be too slow. It's great to prove him wrong! This should be published right away.

A few minor issues resolvable by some text tweaking: the manuscript could use a bit more clarification and detail to better explain how the authors arrived at Puf118 as their protein of interest. There seem to be some logical leaps involving yeast, and that may be the sum of it, but if there is some more directed investigation, that would be informative in the supplement. For instance, the authors find that Puf118 is most similar to yeast Puf4 but the methods by which they made this comparison are vague. Was this based on the nucleotide or amino acid sequence? It would also be helpful if the authors provided more details on how they performed their search for zipcodes / binding motifs? Was it a manual search or a brute force approach? How many localization sequences were tested (apart from the zipcodes and PBEs)? The PUF motif has rather low complexity, it would be expected to be common throughout the genome. The genome wide search that was performed in supplementary figure 1 compares genes related to mitochondria, metabolism and chemotaxis. Were these the three most enriched categories? Is RNA localization and chemotaxis the major function of Puf118? It may be helpful to add the sequence homologies for each of the five Dicty Puf genes into figure S2. Figure S2 also begs the question about why the remaining four Dicty Puf genes are nuclear (as mentioned in the text) and why only two of the four remaining genes were shown in the figure. How similar are these other genes compared to Puf118 and can these differences hint at possible unique functions? Do they also have homologies to Yeast Puf proteins? The paper relies on an initial assumption that yeast Puf4 and Puf118 have the same consensus motif but this was not demonstrated directly.

We appreciate these comments, and all these points have been addressed by text additions.

The connection between shallow/steep gradient in terms of normal Dicty migration is not totally clear. What biological function does the shallow vs steep gradient represent?

The steep gradients used in this study are comparable to most chemotaxis studies in *Dictyostelium*, where micro-needles were used to create an external point source for a cAMP gradient. The shallow gradients are more physiologically realistic and are comparable to gradients a *Dictyostelium* cell would encounter in nature, which we used in the mRNA and protein localization studies. That is why we believe that the difference between steep and shallow gradients is informative. This has been made clearer in the text.

The authors show that RasC (the upstream regulator of the 4 signaling pathways) protein and RNA does not localize in the shallow cAMP gradient. Was this also tested in a steep cAMP gradient also? Could the localization in a steep gradient be Puf118

independent or does this occur through a separate mechanism? This is important to show because Dicty is able to rapidly change its localization direction. The mechanism that the authors propose with Puf118 seems more suitable for long term / persistent migration. There is some similarity to fibroblasts, a non-polarized version lacking ZBP1 can be induced to polarize in a strong chemotactic gradient (Lapidus et al., ZBP1 enhances cell polarity and reduces chemotaxis, PMC4956933). The referencing is a bit sketchy in covering the field. I suggest that they consider referencing the following publication that relates mRNA localization to motility by real time image analysis: "An unbiased analysis method to quantify mRNA localization reveals its correlation with cell motility" Park, et al. PMID:22832165.

More references have been added accordingly.

Finally, it may be worthwhile to relate these findings back to the larger picture. The model that the authors provide is helpful but do the authors expect this to be conserved in other organisms? One example that comes to mind is neutrophil chemotaxis, is there any evidence for a similar mechanism in human cells?

This has been added to the discussion.

The work shows quite clearly that the role of mRNA localization is to bring together many components of the motility mechanism. This is an important concept that we have argued functions with the actin based motility of lamellipods, in response to criticism that synthesis of actin is insufficient to explain protrusion. We posited that it's not only actin but all the many migration related proteins that together are responsible for directed migration. This work gives experimental credence for this concept. I think it should be emphasized in the discussion.

This has been added to the discussion.

In conclusion this manuscript will have broad appeal to investigators many fields including researchers studying Dicty, chemotaxis, pumilio binding proteins and RNA localization.

Sorry I missed your talk at Einstein. This makes up for it!
Rob Singer

Reviewers' Comments:

Reviewer #1:

Remarks to the Author:

The manuscript has been improved strongly. However, I am still very concerned about the molecular mechanisms behind the observed Front/Rear distribution of signaling molecules.

1. Log(F/R), original point 1

It is good to see that data are converted to log(F/R) ratio. Apparently I was not sufficiently clear asking for "present the data on a Log(F/R) scale" . Meant was to show the F/R data on a log scale, which is identical to presenting the log(F/R) data on a linear scale. So the data are presented as asked for. Please delete the statement "since negative values cannot be shown on a log scale, the data are presented on a linear scale.", because this is very confusing.

2. Front localizations (original point 4)

The explanation of the Puf118-dependent and (+)PBE-dependent localization of signaling proteins in the front is not sufficient. It is mentioned "The mechanism involved in the retention of locally translated chemotaxis pathway proteins at the cell front is unknown. A possibility that proteins remain associated with the Puf118 that bound their mRNA seems unlikely since RNA-binding proteins are usually absent from their mRNAs once the translation machinery is loaded 23."

This is a great paper with tantalizing experiments, but the conclusion does not make much sense, unless it explains why (+)PBE-mRNA/protein support chemotaxis and (+)PBE-derived signaling proteins localize to the front and (-)PBE-derived proteins do not. It is mentioned "The mechanism involved in the retention of locally translated chemotaxis pathway proteins at the cell front is unknown. A possibility that proteins remain associated with the Puf118 that bound their mRNA seems unlikely since RNA-binding proteins are usually absent from their mRNAs once the translation machinery is loaded 23." This non-explanation is not sufficient.

For me to understand these experiments I made a small logic tree. The observation is that (+)PBE-mRNA for signaling proteins support chemotaxis, and (+)PBE-mRNA and its encoded (+)PBE-protein are localized in the front. The logic tree is 1) chemotaxis is supported by (+)PBE-mRNA, or 2) chemotaxis is supported by the (+)PBE-protein. I get the impression that the authors do not favor option 1), (+)PBE-mRNA, but this has not been excluded. The authors could discriminate between the role of (+)PBE-mRNA and (+)PBE-protein by expressing mutant (+)PBE-mRNA that does not encode for (functional) protein.

And for option 2, (+)PBE-protein, the logic tree follows with 2a) the proteins of (+)PBE and (-)PBE-transcribed proteins are different in for instance structure, covalent modification or permanent association with another protein, or 2b) (+)PBE and (-)PBE-transcribed proteins are identical. In case 2a) proteins could have a front-mark that keeps them at the front where they were synthesized, and the mark brings them to a new front where they were not synthesized, and this can occur during the entire life of the protein. In case 2b)with identical(+)PBE and (-)PBE proteins, the (+)PBE proteins can localize for some time to the Front where they were synthesized, but after they have disappeared from the front in cytochalasin D, the (+)PBE proteins are probably no longer different from (-)PBE proteins.

To discriminate between 2A and 2B the following:

Proteins have a life time of several hours, whereas the Front and Rear of the cell are formed and reformed on a second to minute time scale. So during the life time of the protein many new fronts are made to which the (+)PBE-derived proteins are expected to associate while the (-)PBE-derived proteins do not. This should be investigated for (+)PBE-derived signaling proteins (e.g. (+) PLA2-GFP with (-)PLA2-GFP as control).

This can be analyzed using two experimental conditions: First, in (+)PLA2-GFP expressing cells in buffer that make a sharp turns by extending a pseudopod in a new direction and retracting the old

pseudopod. Is (+)PLA2-GFP translocating to the new front? It can also be investigated by treating the cells transiently with drugs that inhibit F-actin and retract the front; upon washing out the drug cells make a new Front that has no physiological/biochemical connection with the old front before drug treatment. Before the drug (+) PLA2-GFP is in the front, upon drug treatment (+)PLA2-GFP is not at a specific side of the cells. Where is (+)PLA2-GFP after washing out the drug? Two possibilities a) In the new front. Then, what does (+)PLA2-GFP recognize? F-actin or and F-actin binding protein such as Tub118? And why does (-)PLA2-GFP not recognizes the new front? Possibility b) after washout old (+)PLA2-GFP is not specifically localized at the new front. Then why has (+)PLA2-GFP lost its localization? This experiment is similar to the cytochalasin D treatment of Tub118-GFP expressing cells (Fig 2C), but now with e.g. (+)PLA2-GFP expressing cells showing cell images and log(F/R) data.

Reviewer #2:

Remarks to the Author:

The revised manuscript adequately addresses the points I raised in the previous review.

Reviewer #3:

Remarks to the Author:

With the extensive revisions, the manuscript is improved and will be a key publication in the field of mRNA's role in cell motility and polarity.

REBUTTAL

We have added the *verbatim* comments of Reviewer #1 in black font, and our response are in red font.

1. Log(F/R), original point 1

It is good to see that data are converted to log(F/R) ratio. Apparently I was not sufficiently clear asking for “present the data on a Log(F/R) scale” . Meant was to show the F/R data on a log scale, which is identical to presenting the log(F/R) data on a linear scale. So the data are presented as asked for. Please delete the statement “since negative values cannot be shown on a log scale, the data are presented on a linear scale.”, because this is very confusing.

We had responded to this reviewer’s comment from the first round of reviews in which s/he stated: “*The authors should convert all primary F/R data to log(F/R) value and then take the mean and SD, and present the data on a Log(F/R) scale.*” We converted all the data, as requested, but noted in our response that we do not present the data on a log scale, since negative values cannot be shown on a log scale. We felt we needed to justify this in the text. Nevertheless, given the reviewers’ comment in the current review, we have removed this sentence from the text.

2. Front localizations (original point 4)

The explanation of the Puf118-dependent and (+)PBE-dependent localization of signaling proteins in the front is not sufficient. It is mentioned “The mechanism involved in the retention of locally translated chemotaxis pathway proteins at the cell front is unknown. A possibility that proteins remain associated with the Puf118 that bound their mRNA seems unlikely since RNA-binding proteins are usually absent from their mRNAs once the translation machinery is loaded 23.”

This is a great paper with tantalizing experiments, but the conclusion does not make much sense, unless it explains why (+)PBE-mRNA/protein support chemotaxis and (+)PBE-derived signaling proteins localize to the front and (-)PBE-derived proteins do not. It is mentioned “The mechanism involved in the retention of locally translated chemotaxis pathway proteins at the cell front is unknown. A possibility that proteins remain associated with the Puf118 that bound their mRNA seems unlikely since RNA-binding proteins are usually absent from their mRNAs once the translation machinery is loaded 23.” This non-explanation is not sufficient.

For me to understand these experiments I made a small logic tree. The observation is that (+)PBE-mRNA for signaling proteins support chemotaxis, and (+)PBE-mRNA and its encoded (+)PBE-protein are localized in the front. The logic tree is 1) chemotaxis is supported by (+)PBE-mRNA, or 2) chemotaxis is supported by the (+)PBE-protein. I get the impression that the authors do not favor option 1), (+)PBE-mRNA, but this has not been excluded. The authors could discriminate between the role of (+)PBE-mRNA and (+)PBE-protein by expressing mutant (+)PBE-mRNA that does not encode for (functional) protein.

We thank the reviewer for his/her thoughtful ideas about our work. However, the reviewer is incorrect that we favor Option #2, and not #1. Our data show that BOTH are important, and this is the conclusion that we draw. Obviously, the polarized location of proteins is required for pathway function in proper chemotaxis (Option #2), but their localization is specified by mRNA and Puf18 distribution (Option #1). While the reviewer

constructs a logic tree from which he/she proposes some experiments, the data supporting our conclusions and the resolution of the reviewer's Options are shown in the current data. We may not have pulled the data together to address this point explicitly, but we have now added a paragraph towards the end of the description of the Results to summarize this point. The most relevant experiments in support of our conclusion are:

The importance of protein localization at the cell front: *pla2/sgcA*-null cells do not migrate in a chemotaxis gradient (Fig. 3). Over-expression of (+)PBE Pla2-GFP rescues this mutation phenotype (Fig. 3C), and we know that (+)PBE Pla2-GFP mRNA and Pla2-GFP translated from (+)PBE Pla2-GFP mRNA both localize to the front of the cell (Fig. 1A). Over-expression of (-)PBE Pla2-GFP does not rescue this mutant phenotype (Fig. 3C), and we know that (-)PBE Pla2-GFP mRNA and Pla2-GFP translated from (-)PBE Pla2-GFP mRNA are both diffusely distributed in the cytoplasm (Fig. 3A). Since the coding sequence of the mRNAs is the same in the (-)PBE and (+)PBE mRNAs, this result strongly suggests that localization of these proteins at the front of the cell depends on mRNA localization at that site. **CONCLUSION:** localization of chemotaxis protein at the front of the cell is required for proper chemotaxis, and protein localization is dependent on mRNA localization at the front (i.e., diffuse mRNA localization gives rise to diffuse protein localization, which does not function in the chemotaxis pathway established at the front of the cell).

The importance of mRNA localization at the cell front: PI3K, Pla2, SgcA and TorC/Lst8 act synergistically in chemotaxis (as shown in the *pla2/sgcA*-null cells, for example). Therefore, perturbation of one component will not have a phenotypic effect on chemotaxis. We solved this problem by localizing Puf118, and hence mRNAs for PI3K, Pla2, SgcA and TorC/Lst8, at the rear of the cell using a TalA fusion (Fig. S7); this did not effect mRNA or protein levels of chemotaxis pathways and Act1 (Fig. S5). Significantly, mRNAs of Lst8, PikF, Pla2, SgcA-N and Act1 all co-localized with TalA-Puf118 at the cell rear in a PBE-dependent manner (Fig. 4A, Fig. S7). TalA-Puf118 cells were strongly impaired in their ability to enter (Fig. 4C), and migrate efficiently within (Fig. 4D and S8B) a shallow cAMP gradient (0.1 $\mu\text{M}/\text{mm}$ cAMP) in chemotactic chambers. **CONCLUSION:** proper chemotaxis requires the localization of chemotaxis pathway mRNAs at the cell front, which generates the local accumulation of translated proteins at the cell front.

We have added a short paragraph at the end of the Results part to make these points more clearly (line 148-155):

“In summary, the rescue experiments of *pla2/sgcA*-null cells with either (+)PBE or (-)PBE Pla2-GFP (Fig. 3), and the effects of expressing TalA-Puf118 at the cell rear (Fig. 4) show that proper chemotaxis requires the polarized concentration of PI3K, Pla2, SgcA and TorC/Lst8 at the cell front which dependent on Puf118-bound mRNA localization. Significantly, diffuse distribution of (-)PBE Pla2-GFP mRNA gives rise to diffuse Pla2-GFP protein throughout the cytoplasm; although the Pla2-GFP protein translated from the (-)PBE Pla2-GFP mRNA is the same as that translated from the (+)PBE Pla2-GFP mRNA, it does not function in the chemotaxis pathway because it is not concentrated at the cell front.”

And for option 2, (+)PBE-protein, the logic tree follows with 2a) the proteins of (+)PBE and (-)PBE-transcribed proteins are different in for instance structure, covalent modification or permanent association with another protein, or 2b) (+)PBE and (-)PBE-

transcribed proteins are identical. In case 2a) proteins could have a front-mark that keeps them at the front where they were synthesized, and the mark brings them to a new front where they were not synthesized, and this can occur during the entire life of the protein. In case 2b) with identical (+)PBE and (-)PBE proteins, the (+)PBE proteins can localize for some time to the Front where they were synthesized, but after they have disappeared from the front in cytochalasin D, the (+)PBE proteins are probably no longer different from (-)PBE proteins.

We did not make alterations to the coding sequence of the mRNAs. We made a 2 base mutation in the PBE (to generate the (-)PBE mutants) in the 3' UTR, which does not affect the coding sequence of the mRNA. Thus the conjecture that this somehow affects the "*structure, covalent modification or permanent association with another protein*" encoded by the (+)PBE and (-)PBE mRNAs is incorrect.

To discriminate between 2A and 2B the following:

Proteins have a life time of several hours, whereas the Front and Rear of the cell are formed and reformed on a second to minute time scale. So during the life time of the protein many new fronts are made to which the (+)PBE-derived proteins are expected to associate while the (-)PBE-derived proteins do not. This should be investigated for (+)PBE-derived signaling proteins (e.g. (+) PLA2-GFP with (-)PLA2-GFP as control). This can be analyzed using two experimental conditions: First, in (+)PLA2-GFP expressing cells in buffer that make a sharp turns by extending a pseudopod in a new direction and retracting the old pseudopod. Is (+)PLA2-GFP translocating to the new front?

We report this experiment in Fig S1; another experiment that this reviewer reasonably requested in the first round of reviews. The data show that during dynamic pseudopod movements, (+)PBE Pla2-GFP remains at the cell front. It is meaningless to repeat this experiment with the (-)PBE Pla2-GFP mutant, as this protein is diffusely localized in the cytoplasm and not polarized at the cell front (Fig. 3A).

In addition, concerning the suggestion to look at '*sharp turns*' which can only be induced artificially with a cAMP point source, we noted in our response to the first round of reviews: "We wanted to examine cells responding physiologically to a natural chemotaxis gradient. Thus, we performed all localization studies in natural chemotaxis gradients, i.e. gradients formed by the cells as a population, rather than from an artificial external point source of a very high concentration of cAMP, for example. Since the gradients generated by cells are natural and many co-existing cAMP sources create a dynamic environment, the cells are continuously exposed to subtle changes in gradient direction and, hence reorientation of the cell front; this can be seen in the still images in Fig. 2B. We observed that the polarization distributions of mRNA and proteins at the cell front were indeed dynamic and occurred in a Puf118-dependent fashion, indicating that this regulatory mechanism is indeed within the time-scale of dynamic cell migration that occurs in a natural chemotaxis gradient." Therefore, we will not perform experiments with an artificial point source of high chemoattractant concentration.

It can also be investigated by treating the cells transiently with drugs that inhibit F-actin and retract the front; upon washing out the drug cells make a new Front that has no physiological/biochemical connection with the old front before drug treatment. Before the drug (+) PLA2-GFP is in the front, upon drug treatment (+)PLA2-GFP is not at a specific side of the cells. Where is (+)PLA2-GFP after washing out the drug? Two possibilities a)

In the new front.

As noted in the response above, we have performed a detailed analysis of this point in normal cells, with convincing results, and there is no reason to repeat this in an artificial experiment that uses drug treatment that may affect many aspects of cell organization.

Then, what does (+)PLA2-GFP recognize? F-actin or and F-actin binding protein such as Tub118? And why does (-)PLA2-GFP not recognizes the new front? Possibility b) after washout old (+)PLA2-GFP is not specifically localized at the new front. Then why has (+)PLA2-GFP lost its localization?

As we noted in the rebuttal and revised manuscript, how these chemotaxis pathway proteins are retained at the front of the cell after their translation is unknown – we stated this explicitly in the revised text. We note that it took many different laboratories, and many years of work to define how actin-associated proteins are bound and localized to the plasma membrane and with actin.

It is extremely unfair to ask us to define the mechanism for the localization of these 4 proteins. We have presented evidence that the assembly state of actin is likely important, and that localized actin synthesis at the cell front are critical. Whether this is due to direct binding (unlikely due to the lack of cognate actin binding sites), perhaps a phase transition event, or something else is a completely new project involving years of work. In that context, we want to emphasize that this work defines the critical role of mRNA localization of 4 chemotaxis mRNAs in chemotaxis; this has not been done before, and is highly novel. As reviewer #3 noted, “..... the manuscript will be a **key publication in the field of mRNA's role in cell motility and polarity**”. This reviewer is Robert Singer (he identified himself in the review), and is the pioneer and authority in this field.

Reviewers' Comments:

Reviewer #1:

Remarks to the Author:

The main point of my remarks concern the molecular mechanisms behind the observation from the perspective of the function and localization of the encoded proteins, which is my expertise. I have no concern or specific remarks on localization and function of mRNA; this part of the work is excellent and understood well, which is also appreciated by the other reviewers. The work on the localization of the encoded proteins is done very well, but in my opinion is not sufficiently investigated, explained and discussed. I have taken the attitude to assist in improving the manuscript on these points from the perspective of protein function. In essence the point is: how can proteins that are produced in the front from (+)PBE-mRNA be different from presumed identical proteins produced from (-)PBE-mRNA? Proteins have a life time of a few hours, while the front changes in a second to minute time scale. So repeatedly during the lifetime of the protein a specific front is retracted, the protein delocalizes and a new front is made elsewhere in the cell. The observations suggest that the (+)PBE-mRNA –encoded protein relocates to this new front, while the (-)PBE-mRNA –encoded protein does not. The authors in their reply and manuscript at lines 153 to 155 explain: “although the Pla2-GFP protein translated from the (-)PBE Pla2-GFP mRNA is the same as that translated from the (+)PBE Pla2-GFP mRNA, it does not function in the chemotaxis pathway because it is not concentrated at the cell front” . I fully agree with the statement that the reason for not functioning in chemotaxis is the lack of front localization. However, the real unanswered question is how can identical proteins behave different? This looks like magic and requires a convincing explanation in future times; in this paper this issue needs a full discussion, and where possible experiments to dissect different possibilities. Therefore I made the logic tree: 1) (+)PBE-mRNA is sufficient, or 2) (+)PBE-mRNA and (+)PBE-mRNA –encoded protein are both needed. And subsequently 2A) (+)PBE-mRNA –encoded protein is different from (-)PBE-mRNA –encoded protein (and therefore can relocate to a new front), or 2B) proteins are identical.

Concerning logic step 1, the answer to this point is not correct. The experiments are either (+)PBE-mRNA (in front with localization of encoded protein in front) or (-)PBE-mRNA (diffuse with diffuse localization of encoded protein). The authors never dissected localization of mRNA and protein. The suggested experiment was to investigate whether localization of (+)PBE-mRNA is sufficient, thus (+)PBE-mRNA in front and no protein expressed due to missense mutation in encoding region.

Concerning logic step 2, identical or different proteins the following: Since the coding sequence is identical, different does not mean a different amino acid sequence, but a semi-permanent modification that is made because the protein is synthesized at Tub118. I suggested some possibilities (protein structure, protein complex, acylation or other covalent modifications). It would be extremely exciting and unprecedented to show that (+)PBE-mRNA –encoded proteins are fundamentally different from (-)PBE-mRNA –encoded protein. The discussion above is wrapped together with the question does (+)PBE-mRNA –encoded protein in the cytosol behave different from (-)PBE-mRNA –encoded protein in the cytosol when the cell makes a front. Therefore, I asked for an experiment that explicitly investigates the relocation of (+)PBE-mRNA –encoded protein to a new front. I made several suggestions how to do such experiments, all need to investigate the relocation of (+)PBE-mRNA–encoded protein from the cytosol to a new front. The experiment of Figure S1 shows the localization during migration, but in these experiments no new front is formed. Cells in buffer occasionally make a lateral pseudopod in a new direction at a place that was devoid of F-actin and protrusions for some time (Insall et al); The movie used for figure S1 is likely to contain such events. In my opinion the best experiment is an extension of the experiments presented in figure 2C, showing that cytochalasin D treatment leads to the retraction of fronts and delocalization of Puf118-GFP; after washing out cytochalasin D, cells make new F-actin rich fronts that are enriched in Puf118-GFP. The requested experiment is to investigate the localization of (+)PBE-mRNA and especially the (+)PBE-mRNA –encoded protein after washing out

CD. Does it accumulate to the new front? If it does, while (-)PBE-mRNA -encoded protein obviously does not, it must mean that (+)PBE-mRNA -encoded protein is different from (-)PBE-mRNA -encoded protein. What is different remains unknown, but the experiment will set the stage for future experiments.

I appreciate the importance of mRNA localization of four chemotaxis proteins for chemotaxis. However, unless it is shown that mRNA localization is sufficient (logic point 1), the localization of encoded proteins that must explain the results. I am surprised to read that the authors refuse to do experiments that are feasible and investigate the essence of the molecular mechanism of this exciting and important study.

REBUTTAL

Reviewer #1's critique is appended *verbatim* below, and our responses follow in red.

As before, we sincerely appreciate the time and thought that this reviewer has given to our work, and the suggestion for additional experiments. We have not, *per se*, "refused to do these experiments", but rather disagreed with the rationale for doing them; we note that other suggested experiments were completed in the revised manuscript. As detailed below, we feel strongly that some of these suggested experiments would only provide results and conclusions that duplicate experiments that we have already performed and, therefore, not provide a significantly advance that justifies the work and cost. Other experiments to define how chemotaxis proteins are localized are certainly important, but would require significant time and effort and are beyond the focus of the current work.

REVIEWERS' COMMENTS:

Reviewer #1 (Remarks to the Author):

The main point of my remarks concern the molecular mechanisms behind the observation from the perspective of the function and localization of the encoded proteins, which is my expertise. I have no concern or specific remarks on localization and function of mRNA; this part of the work is excellent and understood well, which is also appreciated by the other reviewers.

We appreciate the strong statement by the reviewer about the importance of chemotaxis pathway mRNA localization (by Puf118) and that the work is excellent and understood well; as we stated before, this was the goal of the current work, which was also identified by the other 2 reviewers.

The work on the localization of the encoded proteins is done very well, but in my opinion is not sufficiently investigated, explained and discussed. I have taken the attitude to assist in improving the manuscript on these points from the perspective of protein function. In essence the point is: how can proteins that are produced in the front from (+)PBE-mRNA be different from presumed identical proteins produced from (-)PBE-mRNA? Proteins have a life time of a few hours, while the front changes in a second to minute time scale. So repeatedly during the lifetime of the protein a specific front is retracted, the protein delocalizes and a new front is made elsewhere in the cell. The observations suggest that the (+)PBE-mRNA –encoded protein relocates to this new front, while the (-)PBE-mRNA –encoded protein does not. The authors in their reply and manuscript at lines 153 to 155 explain: "although the Pla2-GFP protein translated from the (-)PBE Pla2-GFP mRNA is the same as that translated from the (+)PBE Pla2-GFP mRNA, it does not function in the chemotaxis pathway because it is not concentrated at the cell front" . I fully agree with the statement that the reason for not functioning in chemotaxis is the lack of front localization. However, the real unanswered question is how can identical proteins behave different? This looks like magic and requires a convincing explanation in future times; in this paper this issue needs a full discussion, and where possible experiments to dissect different possibilities. Therefore I made the logic tree: 1) (+)PBE-mRNA is sufficient, or 2) (+)PBE-mRNA and (+)PBE-mRNA – encoded protein are both needed. And subsequently 2A) (+)PBE-mRNA –encoded protein is different from (-)PBE-mRNA –encoded protein (and therefore can relocate to a new front), or 2B) proteins are identical.

Concerning logic step 1, the answer to this point is not correct. The experiments are either (+)PBE-mRNA (in front with localization of encoded protein in front) or (-)PBE-mRNA (diffuse with diffuse localization of encoded protein). The authors never dissected localization of mRNA and protein. The suggested experiment was to investigate whether localization of (+)PBE-mRNA is sufficient, thus (+)PBE-mRNA in front and no protein expressed due to missense mutation in encoding region.

This is not a simple undertaking as suggested by the reviewer. We do not know the mechanism for localizing chemotaxis pathway proteins to the cell front, and hence we do not know of a missense mutation in the coding region that would affect protein localization. Of course, a systematic series of mutations of each of the 4 pathway proteins could be undertaken, but this would be extremely time consuming and beyond the scope of the present work. In addition, a confounding problem with this experiment is that some mutations might affect protein function but not localization, which might further complicate conclusions from this type of screen.

Concerning logic step 2, identical or different proteins the following: Since the coding sequence is identical, different does not mean a different amino acid sequence, but a semi-permanent modification that is made because the protein is synthesized at Tub118. I suggested some possibilities (protein structure, protein complex, acylation or other covalent modifications). It would be extremely exciting and unprecedented to show that (+)PBE-mRNA –encoded proteins are fundamentally different from (-)PBE-mRNA – encoded protein. The discussion above is wrapped together with the question does (+)PBE-mRNA –encoded protein in the cytosol behave different from (-)PBE-mRNA – encoded protein in the cytosol when the cell makes a front.

Therefore, I asked for an experiment that explicitly investigates the relocation of (+)PBE-mRNA –encoded protein to a new front. I made several suggestions how to do such experiments, all need to investigate the relocation of (+)PBE-mRNA–encoded protein from the cytosol to a new front. The experiment of Figure S1 shows the localization during migration, but in these experiments no new front is formed. Cells in buffer occasionally make a lateral pseudopod in a new direction at a place that was devoid of F-actin and protrusions for some time (Insall et al); The movie used for figure S1 is likely to contain such events. In my opinion the best experiment is an extension of the experiments presented in figure 2C, showing that cytochalasin D treatment leads to the retraction of fronts and delocalization of Puf118-GFP; after washing out cytochalasin D, cells make new F-actin rich fronts that are enriched in Puf118-GFP. The requested experiment is to investigate the localization of (+)PBE-mRNA and especially the (+)PBE-mRNA –encoded protein after washing out CD. Does it accumulate to the new front? If it does, while (-)PBE-mRNA –encoded protein obviously does not, it must mean that (+)PBE-mRNA –encoded protein is different from (-)PBE-mRNA –encoded protein. What is different remains unknown, but the experiment will set the stage for future experiments.

We really do understand the point that the reviewer is making here, but we argue that the experiments that we have performed address the issue of mRNA and protein localization to newly forming pseudopods. In response to the reviewer's suggestion in the previous round of reviews, we did go back and analyze movies of pseudopod dynamics in cells migrating in a shallow, physiological gradient (Fig. S1); the kymographs show clearly that pseudopod dynamics (retraction, protrusion) are mirrored by changes in Pla2 and Lst8 distributions at the cell front. We feel strongly that these

dynamics under normal conditions are more significant than under conditions of drug-induced actin perturbation suggested by the reviewer. We note also, that the CD washout experiment is somewhat complicated by cell spreading after CD-induced cell rounding, although we have mitigated this affect to some extent with low doses of CD used in the experiment (Fig. 2C).

I appreciate the importance of mRNA localization of four chemotaxis proteins for chemotaxis. However, unless is it shown that mRNA localization is sufficient (logic point 1), the localization of encoded proteins that must explain the results. I am surprised to read that the authors refuse to do experiments that are feasible and investigate the essence of the molecular mechanism of this exciting and important study.

We understand the reviewer's frustration that it seems to that we have ignored his/her suggestions. As we have pointed out, we simply disagree with the reviewer and provide a rationale for why. Nevertheless, we want to come to some common ground so that our and the reviewer's ideas are covered. To do this, we have discussed these points in more detail in the first 2 paragraphs of the Discussion (lines 211-232), which highlight the importance of first mRNA localization at the cell front, and then the subsequent retention of translated proteins there too:

“The goal of this study was to investigate whether there is a common mechanism for coordinating the localization of four chemotaxis pathways (PI3K, Pla2, SgcA and TorC/Lst8) at the cell front for directed cell migration. Our results showed that proper chemotaxis in a shallow, physiological chemoattractant gradient requires Puf118-dependent colocalization of all four chemotaxis pathway mRNAs at the cell front, since expressing (+)PBE, but not (-)PBE Pla2-GFP rescued chemotaxis of pla2/sgcA-null cells (Fig. 3), and TalA-Puf118 mislocalization of chemotactic pathway mRNAs at the cell rear (Fig. 4) inhibited chemotaxis.

However, not only is the localization of chemotaxis pathway mRNAs at the cell front required for chemotaxis, but the subsequent maintenance of PI3K, Pla2, SgcA and TorC/Lst8 protein localization there is also required. For example, the diffuse distribution of (-)PBE Pla2-GFP mRNA and correspondingly Pla2-GFP protein did not rescue chemotaxis in pla2/sgcA-null cells (Fig. 3). This indicates that although the Pla2-GFP protein translated from the (-)PBE Pla2-GFP mRNA is the same as that translated from the (+)PBE Pla2-GFP mRNA, it does not function in the chemotaxis pathway because it is not concentrated at the cell front. It is likely that chemotaxis pathway proteins are specifically localization after synthesis at the cell front, since GFP protein is diffuse even though (+)PBE GFP mRNA is localized at the cell front (Fig. 1A-C). These chemotaxis pathway proteins may bind to proteins associated with the actin cytoskeleton at the cell front²⁷, undergo localized post-translational modifications at the cell front that increases their binding affinity to those proteins, or form complex protein assemblies in biomolecular condensates²⁸, perhaps mediated by locally high concentrations of F-actin in pseudopods; further studies will be required to test these, and other mechanism(s).”

The first paragraph draws the major (and novel) conclusion of our work – the requirement for chemotaxis pathway mRNA localization by Puf118 at the cell front for directed cell migration in a shallow, physiological chemoattractant gradient.

The second paragraph states that localization (retention) of chemotaxis pathway proteins at the cell front is also required, and our experimental evidence for this

conclusion. As per the reviewer's comments, we suggest 3 possible mechanisms for protein localization, and that further experiments are required to investigate them all. That these mechanisms are unknown at present does not detract from the novelty of our work in which we identify a common mechanism for coordinating 4 chemotaxis pathways in time and space for efficient cell migration.